# *De Novo* Transcriptome Meta-Assembly of the Mixotrophic Freshwater Microalga *Euglena gracilis*

**DOI:** 10.3390/genes12060842

**Published:** 2021-05-29

**Authors:** Javier Cordoba, Emilie Perez, Mick Van Vlierberghe, Amandine R. Bertrand, Valérian Lupo, Pierre Cardol, Denis Baurain

**Affiliations:** 1InBioS—PhytoSYSTEMS, Laboratoire de Génétique et Physiologie des Microalgues, ULiège, B-4000 Liège, Belgium; j.cordoba@outlook.es (J.C.); emilie.perez@alumni.uliege.be (E.P.); pierre.cardol@uliege.be (P.C.); 2InBioS—PhytoSYSTEMS, Unit of Eukaryotic Phylogenomics, ULiège, B-4000 Liège, Belgium; mvanvlierberghe@doct.uliege.be (M.V.V.); amandine.bertrand@doct.uliege.be (A.R.B.); valerian.lupo@doct.uliege.be (V.L.)

**Keywords:** transcriptome assembly, gene expression, transcriptional regulation, ontology network, co-expression network, taxonomic analysis, database contamination, kleptoplastidy

## Abstract

*Euglena gracilis* is a well-known photosynthetic microeukaryote considered as the product of a secondary endosymbiosis between a green alga and a phagotrophic unicellular belonging to the same eukaryotic phylum as the parasitic trypanosomatids. As its nuclear genome has proven difficult to sequence, reliable transcriptomes are important for functional studies. In this work, we assembled a new consensus transcriptome by combining sequencing reads from five independent studies. Based on a detailed comparison with two previously released transcriptomes, our consensus transcriptome appears to be the most complete so far. Remapping the reads on it allowed us to compare the expression of the transcripts across multiple culture conditions at once and to infer a functionally annotated network of co-expressed genes. Although the emergence of meaningful gene clusters indicates that some biological signal lies in gene expression levels, our analyses confirm that gene regulation in euglenozoans is not primarily controlled at the transcriptional level. Regarding the origin of *E. gracilis*, we observe a heavily mixed gene ancestry, as previously reported, and rule out sequence contamination as a possible explanation for these observations. Instead, they indicate that this complex alga has evolved through a convoluted process involving much more than two partners.

## 1. Introduction

*Euglena gracilis* is a secondary green alga that can grow in a wide variety of environments. *E. gracilis* belongs to the euglenids, a monophyletic group of free-living, single-celled flagellates that inhabit aquatic ecosystems. Euglenids are distinguished mainly by their unique type of cell covering, the pellicle. The latter is a complex structure composed of proteinaceous strips covered by a cell membrane and underlain by the microtubule system and the cisternae of the endoplasmic reticulum [1]. Together, euglenids, symbiontids (free-living flagellates living in low-oxygen marine sediments), diplonemids (free-living marine flagellates) and kinetoplastids (free-living and parasitic flagellates, e.g., *Trypanosoma*) form the monophyletic group of Euglenozoa [2,3,4,5]. Euglenids are early diverged members of the *Euglenozoa* and distant relatives to the kinetoplastids [6]. Thus, analysing *E. gracilis* genomic information is a way to approach the evolution of parasitism, due to their common ancestry with kinetoplastids [7,8]. For example, it has been shown that many additional subunits of the mitochondrial respiratory chain previously considered exclusive to kinetoplastids are shared with *E. gracilis*, and therefore cannot be associated with the parasitic lifestyle [9]. Yet, it is worth mentioning that free-living bodonids (e.g., *Bodo saltans*) are better comparators for parasitism [10,11]. The relationship between euglenids and kinetoplastids has been first proposed by T. Cavalier-Smith based on ultrastructural similarities (e.g., “mitochondrial cristae shaped like a flattened disc with a narrow neck”) [12], then supported by other lines of evidence, such as alignments of nuclear rRNA [13], the addition of a leader sequence to nuclear pre-mRNAs [14] and the presence of trypanothione reductase in *E. gracilis*, previously found only in kinetoplastids [15].

*E. gracilis* bears a complex plastid [16], derived from a green alga belonging to *Pyramimonadales*, and acquired by a free living phagotrophic eukaryovorous euglenid ancestor [17,18,19]. As the result of a so-called “secondary” endosymbiosis, this chloroplast is bound by three membranes, whereas primary plastids only have two membranes [20,21]. Whatever the specific event, endosymbiosis is accompanied by massive gene loss and gene transfer from the genome of the symbiont to the nuclear genome of the host (Endosymbiotic Gene Transfer or EGT) [22]. Moreover, there can be gene transfers from sources other than the symbiont giving rise to the observed plastid [Horizontal (or Lateral) Gene Transfer or HGT/LGT], for example, over (more or less cryptic) transient endosymbioses (e.g., “shopping bag” [23,24,25] and “red carpet” [26] hypotheses). Alternatively, HGT can occur in a, possibly ulterior, “non-endosymbiotic context” [27,28] (e.g., “limited transfer window” hypothesis” [29]), because it may be easier to duplicate or recruit a foreign gene for servicing the nascent plastid than to get it from the symbiont itself [30]. In any case, both EGT and HGT have shaped the nuclear genome of photosynthetic euglenids, leading to heavy genetic mosaicism (e.g., [7,31,32]).

Due to its great metabolic flexibility, a large number of culture media and growing conditions have been used to study *E. gracilis* over the past 60 years [33,34,35,36,37]. Commonly, the mineral composition remains similar from one medium to another, but three parameters vary greatly: the pH (which can be acidic or neutral), the source of organic carbon (e.g., acetate, ethanol, and succinate) and the concentration of the carbon source (from 10 mM to more than 150 mM). *E. gracilis* can therefore exploit a variety of organic carbon sources, as well in the dark (heterotrophic conditions) as in the light (mixotrophic conditions), where a high concentration of organic carbon leads to a decrease in photosynthesis by repressing chlorophyll biosynthesis, reflecting the fact that this organism switches between nutritional modes and combines them readily [38,39,40]. *E. gracilis* is also known for its atypical metabolic pathways, some of them producing compounds of commercial interest. In photosynthetic euglenoids, carbon reserves are stored in the cytoplasm in the form of paramylon (β-1,3-glucan), in place of the starch (α-1,4 and α-1,6-glucan) typical of the green line [41,42]. Paramylon can be used to produce bioplastics [43] and, similarly to other β-glucans, has been reported to display some anti-tumoural activity [44]. In anoxic (fermentative) conditions, *E. gracilis* has the unique ability among microalgae to convert paramylon into wax ester compounds suitable for drop-in jet biofuels conversion because of their low freezing point [45,46,47]. *E. gracilis* is also used as a source of dietary supplements (e.g., the most bioactive form of vitamin E, α-tocopherol, is present in *E. gracilis* biomass in a relatively high amount) [48].

Due to its evolutionary and biotechnological interests, *E. gracilis* is the best studied member of the euglenids. Its chloroplast genome (143 kb) was among the first plastid genomes ever sequenced [49], while its tiny mitochondrial genome has been recently resolved [50,51]. To date, few studies have used high throughput sequencing technologies to publish Omics information on *E. gracilis* [7,52,53]. In this respect, attempts to sequence its nuclear genome are also very recent (initially estimated between 1 Gb to 9 Gb; see [54] for a review). These efforts have culminated with the release of a very large (500 Mb) and highly fragmented draft genome, as authors recalled, due to gapped contigs or unknown base representation in half of the genome [7].

In this work, we have assembled a consensus transcriptome taking advantage of the raw read data publicly available, including newly generated transcriptomic libraries, for a total of five different data sources. Our assembly protocol was very thorough, with a special emphasis on potential contaminant sequences, resulting in the most complete transcriptome released to date for *E. gracilis*, according to a systematic comparison with the two other public transcriptomes [7,53]. After functional and taxonomic annotation of the predicted coding sequences, we performed a comparative study of their expression level across a range of culture conditions and studies, which allowed us to build an information-rich network of co-expressed genes. However, these results confirm that transcriptional control is not the primary level of genetic regulation in euglenozoans, while our taxonomic analyses point to highly mixed gene ancestry, compatible with a kleptoplastidic phase of plastid acquisition.

## 2. Materials and Methods

### 2.1. Data Collection

#### 2.1.1. Public Repositories

Searching for public RNA-Seq data for *E. gracilis* in the International Nucleotide Sequence Database Collaboration (INSDC) returned eight studies. We further recovered an additional dataset, produced and submitted to the European Nucleotide Archive (ENA) repositories by ourselves (see Section 2.1.2 for details). Of these nine studies, only five short read datasets (5 experiments/23 samples) that used Illumina technology to analyse whole transcriptomes were exploitable. Among the discarded experiments, PRJEB4713 contained 454 GS FLX Titanium long reads, a size that is difficult to handle by the chosen assembler, while PRJEB21674 only included a single euglenid sample (among 1179), yet labelled as “Euglena sp.”, PRJNA294935 primarily contained mitochondrial sequences, and PRJNA12797 (built out of ESTs) was not accessible from public repositories. At last, PRJDB4781 was not included because our meta-assemblies had been completed by the date of its release (October 2019). The data files from the five retained experiments were downloaded using fastq-dump utility from the SRA Toolkit with -I and --split-file arguments to divide files into forward and reverse paired reads. We also collected the two transcriptome assemblies hitherto available, GEFR01 and GDJR01. The former was encoded under study accession PRJNA298469, which corresponds to experiments B and C, and the latter, which corresponds to experiment D, was encoded as study PRJNA289402. For further details on experimental design or/and samples, see Table 1.

#### 2.1.2. In-House Experiments, Cell Culture and Sequencing

The strain of *E. gracilis* (1224-5/25) was obtained from SAG (Sammlung von Algenkulturen Göttingen, Germany). Cells were cultured in liquid mineral medium tris-minimum-phosphate (TMP) at pH 7.0 and 25 °C, supplemented with a mixture of vitamins (vitamin B1 2·10-2 mM, vitamin B8 10-4 mM and vitamin B12 10-4 mM). In three samples, acetate (60 mM) was added as a carbon source, under different photosynthetic photon flux densities (PPFD, T8 fluorescent neon tubes) (in the dark, at low PPFD (50 μE m^−2^ s^−1^) or at medium PPFD (200 μE m^−2^ s^−1^), while in a fourth sample, acetate was not supplied and light was set to low PPFD (50 μE m^−2^ s^−1^). For each sample, the cells in the exponential phase (1–2 × 10^−6^ cells/mL) were recovered by centrifugation, 10 min at 500 g. Total RNA was extracted with the protocol outlined in [55], then fragmented and retro-transcribed before standardization using the Duplex-Specific Nuclease kit (Evrogen, Russia). Each library was prepared using the Illumina total mRNA kit (Illumina, San Diego, CA, USA) and quantified by qPCR using the KAPA Library Quantification Kit (Roche, Switzerland). Subsequently, samples were sequenced in both reading directions (paired-end 2 × 100 nt) on four separate tracks of a high-speed sequencer Illumina HiSeq 2000, yielding on average ca. 235 million reads per sample. Library preparation, DSN normalization and high-throughput sequencing by Illumina technology were carried out by the GIGA genomics platform (https://www.gigagenomics.uliege.be (accessed on 23 July 2014)). Raw reads have been deposited at the ENA database under the study accession number PRJEB38787 (Table 1).

### 2.2. Data Assembly

A schematic representation of the de novo transcriptome reconstruction and analysis pipeline is given in Figure 1. All computations were performed on a grid computer.

#### 2.2.1. Data Pre-Processing

Every raw read file (run accessions SRR/ERR) was treated as one sample, even if two or more files were replicates of the same experimental condition. Once collected and transformed into fastq files, all samples were treated separately. Raw reads were analysed with FastQC v0.11.6 to assess the quality of the data [56]. PRINSEQ-lite.pl v0.20.4 was used to remove reads that contained more than one ambiguous nucleotide [57]. Then, Trimmomatic v0.32 was used with the following parameters (ILLUMINACLIP: TruSeq3-PE.fa:2:30:10 SLIDINGWINDOW: 4:25 LEADING: 3 TRAILING: 3 MINLEN: 25) to truncate the low quality regions of certain sequences and cut adapters and other Illumina-specific sequences from the reads [58]. Output data was sorted into three different batches as paired, unpaired and singleton reads. Finally, read quality was re-assessed using FastQC, and the resulting plots visually compared to those obtained in the beginning to check the effect of the filtering procedure.

#### 2.2.2. Transcriptome Assembly

Pre-processed reads (paired, unpaired and singleton reads) were assembled per experiment in two steps to yield five transcriptomes, one per experiment. We used Trinity v2.4.0 software [59] for de novo transcriptome assembly. During the first step, samples of each experiment were assembled four times, combining values (one/two) of minimum count for k-mers to be assembled (–min_kmer_cov) with normalization turned off (–no_normalize_reads) or on (default) to provide maximal sensitivity for reconstructing lowly expressed transcripts. In all cases, we used the default parameters with a minimum contig length (–min_contig_length) of 100 nt. Second, to reconstruct one single transcriptome per experiment, the four assembled transcriptome replicates were pooled together with the tr2aacds.pl script (using default parameters) from the EvidentialGene v2016.07.11 software package [60,61].

#### 2.2.3. Transcriptome Decontamination

To ensure the purity of the five transcriptomes, we determined the guanine-cytosine (GC) content distribution across reconstructed transcripts. Furthermore, we explored the potential contamination of the five transcriptomes individually by comparing their transcripts against the NCBI nucleotide database (nt) using BLASTN v2.2.28 [62,63]. We used a conservative approach with an E-value threshold of 1 × 10^−50^ and an identity threshold of 90% to maximize the identification of true matches. The best hit for each query was selected, and the organism name (sscinames) of these top matches were collected, tabulated and quantified. Abundant organisms other than *Euglena* were flagged as putative contaminants. To obtain uncontaminated transcriptomes, the original reads were first aligned to the corresponding genomes (downloaded from Ensembl [64] using Bowtie 2 v2.2.6 in local mode (–local –no-unal)) [65,66]. Reads for which the alignment score exceeded the default minimal value of 20 + 8.0 * ln(L), where L is the read length, were removed. Then, the remaining (i.e., unaligned) reads were assembled again following the procedure described in Section 2.2.2.

#### 2.2.4. Generation of a Consensus Transcriptome

The five resulting transcriptomes (one per experiment) were further combined and analysed with the tr2aacds.pl and evgmrna2tsa2.pl (-onlypubset) scripts from EvidentialGene to select the overall best candidate transcripts. The remaining reconstructed transcripts were discarded because they were classified either as redundant, fragmented or uninformative coding sequences, based on untranslated region (UTR) length, gaps, amino acid quality, and stop and start codon presence. After reducing redundancy, EvidentialGene clustered the best transcripts by groups of likely isoforms using CD-HIT v4.6.8 [67,68] and a similarity threshold of 90% on the amino-acid sequences. Sequences were considered as true isoforms (i.e., representing the same gene) when sharing high-identity (≥98%) exon-sized fragments, as determined with BLASTN v2.2.28 (E-value cut off of 1 × 10^−19^). Transcripts proposed by EvidentialGene as the most representative isoform for each gene were selected for annotation (see Section 2.4 and Section 2.5) and for studying gene expression (Section 2.6, Section 2.7, Section 2.8).

### 2.3. Assessment of Transcriptome Quality

Additional analyses were performed to determine the quality of the assembled transcripts. The same set of analyses was also performed on the two other transcriptomes publicly available (GEFR01 [7] and GDJR01 [53]) for comparison with the present study. First, basic statistics based on the length of transcripts and the number of ORFs were computed. Read representation was determined by mapping back the cleaned reads (see Section 2.2.1) to each of the three transcriptomes with the aligner Bowtie 2 v2.2.6 (–local, –no-unal) as described in [65]. Note that unpaired and singleton reads were excluded from all quality statistics. In parallel, we used two evaluation tools, Detonate v1.11 [69] and TransRate v1.0.3 [70], to get reference-free quality scores for the three transcriptomes.

To check the presence of the spliced leader (SL) sequence [14] in the three public transcriptomes, we used wordmatch from the EMBOSS software package [71] and three length thresholds (12, 14 and 24 nt) found in the literature [52,53]. Matches were only considered when falling at the 5′-end of a transcript, whether in forward or reverse orientation, as transcripts are not oriented in the transcriptomes. More precisely, each transcript was first reverse-complemented, and both versions (forward and reverse) were truncated at 40 nt before running wordmatch. Besides, transcripts actually corresponding to rRNA sequences were identified by combining RNAmmer v1.2 [72] and MegaBLAST v2.2.28 [62] searches (E-value cut-off of 1 × 10^−50^, the latter using accessions X12890.1 (*E. gracilis* rrnC operon), M12677.1 (SSU rRNA 18S) and X53361.2 (LSU rRNA 28S) as queries. Regarding coding sequences, we estimated the numbers of putative genes with GeneMarkS-T (beta version) [73] and measured transcriptome completeness with BUSCO v.3.0.1 [74,75] using both “Eukaryota” and “Protists *ensembl*” datasets.

Lastly, we used CD-HIT-2D v4.6.8 [67,68] to identify similar predicted protein sequences between transcriptomes with our transcriptome as a reference. We explored different word sizes (2 to 5) at several thresholds of sequence identity (ranging from 0.5 to 0.9). Sequences from the other two public transcriptomes that could not be clustered with sequences of our consensus transcriptome were tentatively aligned using BLASTP v2.2.28 instead [62]. We further calculated the expression of presumably “missing” sequences in GDJR01 (D) and GEFR01 (B-C), respectively, following the procedure described in Section 2.5. The sequence was deemed invalid and not considered missing if its expression was below one transcript per kilobase million (TPM) in the transcriptome from which it had been identified. In a complementary analysis, highly similar nucleotide sequences from the three transcriptomes were clustered all together at once using CD-HIT-EST (identity threshold of 0.9, word size of 8, coverage of the shorter sequence of 0.9). Within each cluster, transcripts were pooled per transcriptome and their properties used to compare the three transcriptomes over all clusters, in terms of redundancy, length and identity. Analyses were performed either on all clusters or only on clusters shared across the three transcriptomes.

### 2.4. Transcript Annotation

The annotation procedure was carried out in three steps. First, assembled transcripts (i.e., the EvidentialGene representative isoforms) were annotated with EggNOG-mapper v1 [76,77]. We used HMMER to compare our data with the eukaryotic database of EggNOG, prioritizing coverage. Second, we annotated our transcripts by similarity using PSI-BLAST v2.2.28 searches [62] (E-value cut-off of 0.001) against Swiss-Prot [78]. Third, we aligned the assembled transcripts to the NCBI protein (*nr*) database [63] using TBLASTN v2.2.28 [62] (same E-value cut-off). We recovered Gene Ontology terms (GO) [79] and Kyoto Encyclopedia of Genes and Genomes Orthologs terms (KO) [80] of each transcript for further term enrichment analysis and network representation (see Section 2.7 for details). For that purpose, EggNOG features were assigned when possible to a transcript; if annotation was missing, PSI-BLAST v2.2.28 annotation was provided instead, or even TBLASTN v2.2.28 features whenever the two first previous methods failed. For mitochondrion and plastid-specific analyses, the components of the photosynthetic and respiratory electron transport chains were identified by BLASTP v2.2.28 searches [62] (E-value cut-off of 0.001) against reference proteins described in the literature. Hence, respiratory subunits were taken from [9,81,82], whereas subunits of photosystem I, photosystem II, cytochrome b6f complex, cF1Fo ATP-synthase were sourced from [83], and LHC polyproteins from [84].

### 2.5. Taxonomic Analyses

Taxonomic affinities were determined based on BLASTX v2.2.28 [62] searches against a broadly sampled proteome database, composed of 73 manually selected eukaryotes [85] and 19,802 representative prokaryotes subsampled from a curated database of 27,762 genomes [86]. For each assembled transcript, a last common ancestor (LCA) was computed based on their closest relatives (best hits, if any) in the database, provided they had a bit-score ≥80 and were within 95% of the bit-score of the first hit (MEGAN-like algorithm [86,87]). Organellar (plastid and mitochondrion) encoded proteins were distinguished from nuclear-encoded proteins by querying (BLASTP) two *E. gracilis* organelle databases assembled from the NCBI RefSeq “Proteins” portal [63]. To identify with certainty an organelle-encoded protein, only hits with a minimum percentage identity of 99% and a strictly identical length were considered. Such organelle-encoded sequences were expected at least from our own reads, which were generated in the absence of poly-A selection.

In parallel, tetranucleotide frequencies (TNFs) were computed for individual transcripts using the default settings of compseq from the EMBOSS software package [71]. Then, assembled transcripts for which a taxonomic affiliation had been obtained were ranked following their GC content and split into four partitions of equal size in terms of number of transcripts. Finally, ten principal component analyses (PCAs) were computed on TNFs, each one based on 1000 randomly chosen transcripts, using the prcomp function of the STATS v3.4.3 R base package [88]. For each PCA, two different colour schemes were applied on data points: the broad taxonomic affiliation of the transcript LCA (divided into four groups: Viridiplantae, Kinetoplastida, other Eukaryota and Bacteria), and the GC-content partition of the transcript.

### 2.6. Expression Quantification

The abundance of assembled transcripts was estimated by using RSEM v1.2.31 [89] and Bowtie2 v2.2.6 aligner [65,66]. Specifically, we used the align_and_estimate_abundance.pl Perl script wrapped in the Trinity v2.4.0 software package [59]. Data was then processed with abundance_estimates_to_matrix.pl Perl script without normalization parameters to generate the final expression matrix. Expression values are provided in transcripts per kilobase million (TPM) and pooled per gene (i.e., gene-level counts) [90].

Each count value was log2-transformed and converted to a Z-score to make samples comparable (sample mean was subtracted from each sample observation and divided by sample standard deviation). Batch effects were tentatively removed with the help of the SVA v.3.26.0 R package [91], so as to adjust data for unwanted sources of variation. However, such correction proved to be ineffective and thus abandoned (see Results and Discussion). For downstream analyses, only the 2500 most variable genes were retained (based on their expression variance across the 23 samples).

### 2.7. Gene Clustering Based on Expression Profiles

The 2500 most variable genes were clustered using the Partitioning around medoids (PAM) algorithm (from the CLUSTER v.2.0.7 R package) [92], which creates a fixed number of clusters (k) by minimizing the sum of the dissimilarities of the observations to their closest representative object (medoid). To capture both positive and negative relationships between gene pairs, we used a dissimilarity matrix of expression based on the squared Pearson correlation (d = 1 − r^2^). The optimal cluster segregation was selected by cycling through the number of potential solutions, ranging from k = 5 to 75. In each solution, an average of maximal absolute correlations within-cluster (w-k cor_max_) and an average of minimum absolute correlations between-cluster medoids (b-k cor_min_) were computed. To intercept the point where optimal cluster segregation occurred, a reinterpretation of the Dunn index was used, and we computed the b-k cor_min_ and w-k cor_max_ ratio, choosing the solution with the minimal ratio value. At this optimal point, decreasing or increasing the number of cluster solutions would not better explain the data [93]. Heat map and hierarchical clustering analyses (correlation was used as the distance and centroid linkage clustering as the method) of expression data were carried out using the pheatmap function from the pheatmap v1.0.12 R package [94] and, when necessary, row-wise data (gene expression of the transcripts) was aggregated using k-means clustering to facilitate visual inspection of expression across conditions.

### 2.8. Gene Ontology (Enrichment) Analyses

The clusters based on the 2500 most variable genes were further analysed to visualize overrepresented biological terms using the whole GO and KEGG term space from Section 2.4 as a background. We explored enriched pathways within the expression clusters using ClueGo v2.5.0 tool [95], a visualization plug-in implemented in the Cytoscape v3.6.0 environment [96]. Term overrepresentation was estimated by an enrichment test based on the hypergeometric distribution followed by Benjamini–Hochberg adjustment for multiple testing. An annotation network was built with the ClueGo plug-in from kappa scores, which reflect the associations between genes and GO and KEGG terms. Network specificity was set between 3 and 12 GO hierarchy levels, and term selection was set to a minimum of 3% genes per cluster. Kappa score threshold was set to 0.3, and we allowed GO parent-child term fusion. Moreover, we explored the network with the MCODE algorithm [97], implemented as a Cytoscape plug-in, to detect densely connected regions or hubs in the network. Those hubs were found in the network establishing a degree cut-off of 2 for network scoring criteria, without including loops. Option Fluff was selected and parameters for Cluster Finding panel were set at 0.1 and 0.2 for node density and node score cut-off, respectively, a minimum of 2 edges per node of cluster cores (K-Core) and a maximum depth of 100.

## 3. Results and Discussion

### 3.1. Data Collection/Datasets

Out of the eight datasets publicly available for *E. gracilis*, only four [PRJNA310762 (A), PRJEB10085 (B), PRJNA298469 (C), PRJNA289402 (D)], were retained to assemble our consensus transcriptome, along with our own experiment PRJEB38787 (E; Table 1), which used Duplex-Specific thermostable nuclease (DSN) normalization to avoid poly-A selection. These five datasets totalled circa 2.6 billion raw Illumina reads (100-nt long), of which 70% belong to our experiment. After quality treatment, between 5 and 7% of reads were lost in experiments PRJNA310762 (A), PRJNA298469 (C) and PRJNA289402 (D), whereas the rejection of reads was more important in experiments PRJEB10085 (B) and PRJEB38787 (E). In PRJEB10085 (B), 19% of reads were truncated as a consequence of low-quality regions, whereas in PRJEB38787 (E), 50% of reads were discarded because of the high number of ambiguous nucleotides, especially in reverse reads. Hence, we got 57.8 million of good quality reads out of 62 after pre-processing of experiment PRJNA310762 (A) [7], 310 million reads out of 383 for experiment PRJEB10085 (B) [52], and 267.7 million from experiment PRJNA289402 (D). In the latter case, we used all samples as input, whereas Yoshida et al. (2016) only used the reads from cells grown in mixotrophic conditions to build their assembly [53]. Finally, Ebenezer et al. (2019) used 410 million reads as input for their transcriptome assembly, probably as the result of combining reads from PRJEB10085 (B) and PRJNA298469 [7].

After quality filtering, ca. 1.5 billion reads were retained, pre-processed read files of each individual experiment were assembled in four replicates using Trinity and then condensed into one individual transcriptome per experiment using EvidentialGene, which served as the basis for creating the consensus transcriptome (see Materials and Methods for details). Overall, PRJEB38787 (E), PRJEB10085 (B), PRJNA289402 (D), PRJNA310762 (A) and PRJNA298469 (C) experiments accounted for 55, 20, 17, 4, and 2% of the pre-processed reads used for the individual assemblies, respectively.

### 3.2. De Novo Assembly Evaluation

#### 3.2.1. Individual Assemblies

The presence of sequences within a data set that originate from sources other than the sequenced sample is a known limitation of RNA-Seq experiments (e.g., [98,99] in human datasets). For some studies, such as large-scale phylogenomics, contaminants can be very problematic and must be dealt with using an array of different approaches [100]. Thus, before combining the individual five transcriptomes into a final consensus transcriptome, all assembled sequences were BLASTed against the NCBI nucleotide (*nt*) database [63] to identify possible contaminants. Using stringent thresholds, we found in the five transcriptomes only 948 unique hits of reconstructed transcripts that matched organisms other than *E. gracilis*. These organisms were considered as possible contaminants. Among them, we selected the five organisms whose abundance was the greatest (*Homo sapiens*, *Saccharomyces cerevisiae*, *Escherichia coli*, *Ovis aries* and *Caenorhabditis elegans*). It is noteworthy that sheep (and cow) DNA is commonly sequenced on our genomic platform. By mapping all pre-processed reads to the nuclear genome of these five species, we found that contaminants were less than 0.01% of the reads matching one of the contaminant genomes. In comparison, it has been shown that 0.13% of contaminant reads were present on average in a subset of 150 sequencing data files from the 1000 Genomes Project [101]. In the case of PRJNA298469 (C), we flagged as contaminants 68 reads per million reads (RPM), a larger proportion compared to the other experiments, which varied between 2 and 29 RPM (Table 2). Contaminant reads were removed and new assemblies of each experiment were generated anew from decontaminated reads, following the same procedure as above (see Section 2.2.2 for details). Afterwards, a new BLAST analysis was performed to quantify whether the contamination level was reduced. As expected, hits matching to *C. elegans*, *Escherichia coli*, *H. sapiens*, *O. aries* and *Saccharomyces cerevisiae* decreased, while hits matching to *Euglena* remained similar (Appendix A). Besides, we traced the non-*Euglena* sequences that persisted in the final consensus transcriptome presented just below (see Section 3.2.2). Overall, from 716 unique hits of non-*Euglena* sequences identified with the latter BLAST analysis, only 64 were still present in the final consensus transcriptome (see Section 3.3.2 for details on the contamination sources). As a case in point, the complex genetic makeup of *E. gracilis* (e.g., [52]) makes it difficult to determine when a sequence, even if very peculiar, has been acquired from a very distantly related species or whether it can be a contaminant (see also Section 3.3.2 for an attempt to differentiate the two cases). For example, the glyoxylate cycle is localized within the mitochondria in *E. gracilis* and isocitrate lyase and malate synthase form only one bifunctional enzyme, called EgGCE [102,103]. A bifunctional enzyme for the glyoxylate cycle is also found in the worm *C. elegans* (opisthokonts), revealing an independent acquisition of the bifunctional enzyme by convergent evolution in these two organisms [104].

The five decontaminated individual transcriptomes were then evaluated with TransRate to check their uniformity. Four transcriptomes yielded ca. 42,342 (±6159) transcripts on average, whilst the number of reconstructed sequences in experiment PRJEB10085 (B) was more than twice the average, 95,490 sequences (Table 2). In addition, the computed GC content was 58% for experiment PRJEB10085 (B), a lower percentage compared to the other assembled transcriptomes, which was around 64%. Finally, we discovered a high frequency of sequences under 500 nt and characterized by a lower GC content (Appendix A). After those small sequences were removed (representing 62% of the transcripts), TransRate statistics were recomputed and yielded values more in line with other experiments, both in terms of number of sequences (36,287) and GC content (62%). We could not determine what the removed sequences were by similarity searches. They might represent some sort of artefact, contamination, or even be the result of a specific feature of experiment PRJEB10085 (B), for example the sequencing of a different strain, i.e., *E. gracilis* var. saccharophila Klebs (SAG 1224/7a) [52], whereas the other four experiments all used the Z strain (SAG 1224-5/25).

#### 3.2.2. Final Consensus Transcriptome

To obtain our final transcriptome, we combined the individual five decontaminated transcriptomes into a consensus transcriptome. Regardless of the aforementioned differences in the amount of pre-processed reads per dataset, the contribution of transcripts from each study in the final consensus transcriptome was rather balanced, where PRJEB10085 (B), PRJNA310762 (A), PRJNA289402 (D), PRJEB38787 (E), and PRJNA298469 (C) accounted for 30.3%, 24.3%, 21.5%, 12.7%, and 11.1%, respectively (Table 2). The resulting transcripts were classified into non-redundant protein-encoding genes, and one representative isoform was selected for each gene. Our new transcriptome was then compared with the other two publicly available transcriptomes, GDJR01 (D) [53] and GEFR01 (B-C) [7] (Table 3). Ebenezer et al. (2019) [7] used a combination of in-house generated sequences (PRJNA298469 (C)) and publicly available data from O’Neill et al. (2015) [52] (PRJEB10085 (B)) to assemble a transcriptome. Assembly transcriptome statistics were computed with TransRate. The overall number of sequences reported in the present work is 91,040, with N50 of 1432 nt, whereas in GDJR01 (D), it was 113,152 (N50 1604), and 72,506 (N50 1242) in GEFR01 (B-C). The mean length of our transcripts was 1096 nt, a value closer to GDJR01 (D) than GEFR01 (B-C), which was ca. 200 nt smaller. The number of protein coding regions predicted by GeneMarkS-T (58,542) and the number of open reading frames (ORF) found with TransRate (62,287) are slightly smaller than in GDJR01 (D), but about twice greater than in GEFR01 (B-C). Our own sequences were classified into 49,922 predicted non-redundant protein-encoding genes, which is comparable to GDJR01 (D), but almost eighteen thousand genes more than in GEFR01 (B-C). As expected, these recomputed numbers are similar to those reported in the original publications of Yoshida et al. (2016) [53] and Ebenezer et al. (2019) [7]. Additionally, O’Neill et al. (2015) [52] found over 32,000 unique components for their *E. gracilis* transcriptome. The total size of our consensus transcriptome is 100 Mb, whilst the size of GDJR01 (D) is 122 Mb, 63 Mb for GEFR01 (B-C) and 38.4 Mb for O’Neill et al. (2015) [52] transcriptome. Overall, the genome size of *E. gracilis* has been estimated from total DNA content to range between 1 Gbp to 9 Gbp [54]. In contrast, the most recent estimation based on high throughput sequencing data was 332–500 Mb in size for the whole haploid genome [7] but, because half of the genome is gapped or has unknown base representation, the authors pointed out that this latter estimation was likely to be approximate.

The pre-processed reads from the five experiments were aligned back to the three public transcriptomes as a metric of completeness. In most cases, the percentage of mapping was over 80%, reaching even more than 90%, with the exception of reads produced by ourselves PRJEB38787 (E), which had a representation of ~75% and ~50% in GEFR01 (B-C) and GDJR01 (D), respectively (Table 4). It is probable that our reads have a lower mapping percentage because they were generated from DSN-normalized total RNA samples, for which analyses of a preliminary sequencing lane revealed many reads corresponding to non-mRNA sequences (e.g., rRNA). However, the specifically low mapping to GDJR01 (D) cannot be explained easily because “transcripts” matching to rRNA sequences were identified in all three public transcriptomes (Appendix A).

Using BUSCO on our predicted proteins, we found that the consensus transcriptome contained 84.8% of complete eukaryotic orthologs and half of them were duplicated, while 10.6% were missing (Figure 2). In comparison, we estimated the completeness of GDJR01 (D) at 80.8% of complete orthologs, of which a fifth were duplicated, and completeness of GEFR01 (B-C) at 76.9%, with only 4% of them duplicated. Moreover, we observed that lower percentages of complete orthologs were accompanied by higher numbers of fragmented and missed sequences. Overall, our consensus transcriptome appears to be the most complete, GEFR01 (B-C) being the least. Ebenezer et al. (2019) [7] also determined BUSCO completeness in GDJR01 (D) and GEFR01 (B-C) transcriptomes in addition to the original transcriptome presented by O’Neill et al. (2015) [52] and similarly concluded that GEFR01 (B-C) was the least complete transcriptome. Beyond transcripts missing due to low expression, discrepancies in the number of complete orthologs predicted by the different studies may also be due to the use of different tools for protein prediction. Whereas we used cdna_bestorf.pl script from EvidentialGene, the other studies used TransDecoder [59], which, reportedly, tends to predict larger amounts of proteins, but performs worse for true transcripts [105]. Despite these differences, the general representation scores of the reads in the assembled transcripts were similar across the three public transcriptomes, even if depending on the exact evaluation software used (Table 5).

As already mentioned, one evidence supporting the evolutionary relationship between trypanosomatids and euglenids are trans-splicing mechanisms [14]. We found that the SL-sequence was present in no more than 10.8% of transcripts in our transcriptome, far from the approximately 53–60% prevalence reported before [14,53], and closer to the 16% found by [52]. However, when performing the exact same analysis on the other two public transcriptomes, we find contrasting results, with SL-sequence matches recovered in at most of 2% and 30.3% of GEFR01 and GDJR01, respectively (Table 6). This indicates that the transcriptome of Yoshida et al. (2016) [53] has the most complete transcripts in 5-end, even though our own assembly includes 200 transcripts with a full-length perfect match to the 24-nt SL-sequence (vs. 45 and 5 for GEFR01 and GDJR01, respectively). Comparison of the mapping coverage for the three public transcriptomes shows that partial matches (12–14 nt) are much more numerous than full-length matches, as expected, but that the former are concentrated at the very beginning of the transcripts, which suggests that they are genuine SL-sequences (Appendix A).

Finally, we determined whether sequences of the other two available transcriptomes were present in our consensus transcriptome through two complementary approaches: one pairwise, sensitive and based on protein sequences, and one global, conservative and based on nucleotide sequences (Appendix A). First, when using CD-HIT-2D with our transcriptome as a reference, a word size of 2 and an identity threshold of 0.4, 26.1% (34,490) of total sequences from GDJR01 (D) were missing and 37.6% (28,552) of total sequences from GEFR01 (B-C). Missing sequences were BLASTed (TBLASTN E-value cut-off of 0.001) against our transcriptome, and 20.5% (27,152) of total sequences of GDJR01 (D) were recaptured and 24.8% (18,870) of GEFR01 (B-C) (Appendix A). After computing TPM values using the pre-processed reads generated in this study, we found that only 518 missing sequences of GDJR01 (D) were expressed above 1 TPM and 1595 in GEFR01 (B-C), which means that potentially 0.5% and 2% of the truly expressed sequences from GDJR01 (D) and GEFR01 (B-C), respectively, are missing from our consensus transcriptome. Hence, these sensitive analyses suggest that we captured more than 98% of the sequences produced in the other transcriptomes hitherto published. Second, CD-HIT-EST was used to compute clusters of related transcripts at an identity threshold of 90%. We recovered 121,851 clusters, in which the three transcriptomes had very similar patterns of presence and representation (Appendix A). Hence, each transcriptome had at least one transcript in 60,220 to 66,041 clusters, whereas they each provided the representative (longest) sequence in 39,434 to 41,610 clusters. Singleton cluster statistics were slightly different, with GEFR01 having 29,997 specific clusters, followed by GDJR01 (27,058) and then our own transcriptome (19,028). When focusing on the 24,164 clusters shared between the three transcriptomes, we see that our transcriptome contributes the highest number of representative sequences, which confirms that they are generally longer than their homologues in the other two transcriptomes. This is also visible in a direct comparison of the mean an maximum transcript length across the three transcriptomes, whether on the 121,851 or the 24,164 clusters (Appendix A). In contrast, comparison of the median and max identity between transcripts of the three datasets reveals that GEFR01 sequences are the most similar on average to the sequences from the two other transcriptomes. They are also the less redundant, with the lowest number of transcripts per cluster.

Altogether, these comparative analyses indicate that the three publicly available transcriptomes each have a distinct edge on the other two: Ebenezer et al. (2019) [7] assembled a compact set of sequences nonetheless providing a large fraction of unique transcripts, whereas Yoshida et al. (2016) [53] obtained a more redundant transcriptome, but with many transcripts complete at their 5-end, as evidenced by the detection of SL-sequences, and for our part, we generated the longest transcripts on average, including a few hundred featuring a full-length SL-sequence, with moderate redundancy.

### 3.3. Global (Transcriptome) Annotation

#### 3.3.1. Functional Annotation of Transcripts

The combination of annotation strategies in our 49,922 predicted non-redundant protein-encoding genes yielded 9916 sequences with GO terms, 7775 KEGG orthologs, 13,298 sequences with a functional annotation and 13,850 with a taxonomic affiliation (Appendix A; see also Section 3.3.2). In the same way, O’Neill et al. (2015) [52] found 14,389 proteins with annotated functions out of the 32,128 predicted proteins of their transcriptome, whereas out of the 49,826 unique components reported by Yoshida et al. (2016) [53], approximately 11,314 were functionally annotated. Ebenezer et al. (2019) [7] annotated over 19,000 sequences, but without discerning what kind of attributes were associated in each case.

In comparison to the annotation performed in the other transcriptomes, we were able to find all the enzymes of the mevalonate pathway, including the diphosphomevalonate decarboxylase (EC 4.1.1.33), which was missing in the work of O’Neill et al. (2015) [52], thereby revealing that the last reaction is catalysed by a canonical enzyme. Regarding the carbohydrate-active enzymes, we found results similar to those outlined by O’Neill et al. (2015) [52]. Hence, we identified a great number of glycosyltransferases (311) and glycoside hydrolases (80), of which a quarter (19) were different types of glucanases (Appendix A). Corroborating the results of Yoshida et al. (2016) [53], we found two transcripts encoding glucan synthases, but could not identify transcripts encoding a 1,3-β-D-glucan phosphorylase, despite that such an enzyme has been previously characterised biochemically [106,107].

In *E. gracilis*, the photoreceptor is considered by some authors to be a rhodopsin-like protein where the retinal chromophore is a carotenoid [108]. We found five enzymes involved in retinol metabolism (EC 2.3.1.76; EC 3.1.1.64, EC 2.3.1.135; EC 1.1.1.105, EC 1.3.99.23) but, in line with Ebenezer et al.’s (2019) [7] findings, we could not find any rhodopsin-like protein candidates. Instead, we found 47 genes involved in visual perception processes (GO:0007601) and, more broadly, 333 genes related to photoresponse (Appendix A), including 13 cAMP/cGMP phosphodiesterases involved in amplification of luminous signal, 15 GTPase regulators, nine arrestins, which are important for regulating signal transduction at G protein-coupled receptors, eight cryptochromes, and three cyclic nucleotide-gated channels of rod photoreceptors. In addition, we found 13 proteins of the paraflagellar rod, a structure observed in euglenids, kinetoplastids and dinoflagellates [109,110,111]. Such a structure is associated with the paraflagellar body (also called paraxonemal body, PAB) in *E. gracilis* [112]. We also found 49 transcripts coding for photoactivated adenylate cyclases (PAC), which are light-sensitive proteins of PAB [113]. Of these, 43 clearly show a bacterial affinity in our analyses, whereas two are highly similar two trypanosomatid sequences [114].

To better understand the general functionality of the consensus transcriptome, we reported the GO annotation results as high-level terms of the three ontologies without the detail of the specific fine-grained terms. For such a task, we used the generic GO Slim Mapper tool of The Saccharomyces Genome Database [115], and the list of summarized GO terms (GO slim) can be found in Appendix A. As we used a compendium of culture conditions, we expected to capture the sum of functionalities represented by the studies individually. We found a total number of 164 GO terms after GO slim analysis, represented by core metabolism (41), transport (13), cell organization (15) and maintenance (25), nucleotide metabolism (35) and protein synthesis (17), vesicle or cilium organization (15) among others. The annotation from O’Neill et al. (2015) [52] was classified into 157 GO categories while Yoshida et al. (2016) [53] determined, under mixotrophic conditions, that the main functional categories were genetic information processing (399 components), translation (291 components), and energy metabolism (239 components). Besides, genes belonging to the latter three categories were generally down-regulated during anaerobic treatment [53]. In the same way, Ebenezer et al. (2019) [7] indicated that major categories were dominated by core metabolic, structural and informational process supergroups, consistent with the current work and previous studies [52,53].

#### 3.3.2. Taxonomic Annotation of Transcripts

As a complex alga resulting from a secondary endosymbiosis between a euglenozoan host and a chlorophyte alga, *E. gracilis* bears genes from multiple origins [16,25]. In terms of sequence similarity (and depending on the current sampling in reference organisms), its nuclear genome is expected to be composed of four main gene classes: (i) *Euglena*-specific genes, (ii) kinetoplastid-specific genes, (iii) eukaryotic genes (i.e., widespread in other eukaryotes), and (iv) (green) genes acquired during the secondary endosymbiosis [31]. Over the last fifteen years, this issue has been extensively studied, both using similarity [52,53] and phylogenetic [7,9,31,32,116,117,118,119] approaches, either at small (i.e., targeted subsets) [9,116,117,118] or larger (i.e., transcriptomic) scales and, when at larger scale, either by focusing on the chloroplast [119] or by surveying “unbiased” transcript collections [7,31,32,52,53]. All these studies have revealed that *E. gracilis* display sequence similarities to a panel of organisms that is larger than predicted by a simple theory of secondary symbiogenesis [120,121]. Unsurprisingly, our large-scale similarity analyses of the consensus transcriptome confirm the results of these previous works (Figure 3). A first observation is that only 28% of the predicted non-redundant protein-encoding genes (13,850 out of 49,922) bear any exploitable similarity with sequences in reference databases. Among those, 937 (7%) correspond to organisms to which we could not assign a specific taxon, whereas 4054 (29%) were only identified as “Eukaryota”. The remaining gene similarities are distributed among kinetoplastids (1364, 10%), green plants (977, 7%) and other subgroups of eukaryotes, whether photosynthetic, such as cryptophytes (530, 4%) and haptophytes (468, 3%), or not, e.g., opisthokonts (947, 7%). Bacterial groups account for 1690 transcripts (12%), among which the most prominent are proteobacteria (34% of bacteria) and cyanobacteria (212, 13%). Only 40 (2%) and 15 (0.9%) transcripts are affiliated to the PVC group or Chlamydiae, respectively [122]. As expected [31], focusing on 119 nuclear-encoded genes involved in mitochondrial and photosynthetic electron transfer chains increases the similarity signal in favour of kinetoplastids (20 out of 86, 22%) and green plants (20 out of 33, 58%), respectively (Appendix A; see also HTML Appendix A).

Similarly to other complex algae (e.g., cryptophytes and chlorarachniophytes [123], ochrophytes and haptophytes [124,125]), *E. gracilis* transcriptomes show a heavily mixed ancestry in terms of gene donor lineages. However, it is a known (yet somewhat neglected) issue that publicly available transcriptomes can be contaminated by foreign sequences because of ecology (e.g., predator–prey, host–parasite or symbiotic relationships), or due to cross-contamination (either in the lab or on sequencing platforms) (see [126] and references therein). That is why we exerted special care to avoid including non-*Euglena* transcripts when assembling the five individual transcriptomes (see Section 3.2.1). In our final consensus transcriptome, we still identified 64 sequences as contaminants, of which 23 are false positives, owing to strong sequence similarity with different kinetoplastids (9 transcripts), green plants or algae (7), or non-green microalgae (7). Since the transcriptome had already been publicly released at the time, the other 41 remaining sequences were retained in subsequent analyses, but tagged as contaminants (Appendix A). Moreover, we used the taxonomic annotation of the 13,850 annotated transcripts to determine whether contaminants could be identified by their base composition pattern (see [127] and references therein). To this end, PCA plots were computed based on transcript tetranucleotide frequencies. Two types of colour annotation were then applied: one following a scale of GC-content and one following the taxonomy (Appendix A). It appears that the taxonomic signal is mixed throughout these PCAs, whereas GC-content clearly corresponds to the PC1 axis. Thus, it was not possible in our case to identify and sort out contaminated transcripts (if any) from *Euglena* transcripts with this approach.

### 3.4. Systematic Functional Annotation of Top Differentially Expressed Genes

To better understand the functional organization of the most relevant *E. gracilis* genes under the assayed culture conditions, we computed a network of ontologies, based on transcript expression levels across all samples and studies (Appendix A). For this purpose, we only selected GO and KEEG terms that corresponded to the 2500 most variable genes (in terms of expression) to determine which biological functions were represented and how they were related to each other. The resulting organized network contained 119 nodes, with an average of nine neighbours per node, and 436 genes from the initial 2500 genes were retained (some genes being part of multiple hubs). We then used the MCODE algorithm to find evidence of higher order organization (Figure 4). The network was composed of nine modules (or hubs), each defined by one ontological category (Appendix A). Hub number 1 (72 transcripts) reflects “regulation of DNA damage checkpoint”, with transcripts involved in apoptosis, control of transcription and other developmental processes. Unlike hub number 7 (see below), hub 1 has a stress response component. Hub 2 (191 transcripts) is the largest hub, and comprises genes involved in translational initiation and termination, or protein targeting to a membrane, and is thus defined by “ribosome” terms. Hub 2 is connected to hubs 3, 5 and 6 in the network. Categorized as a “thylakoid” hub, hub 3 (133 transcripts) is the second largest hub. It mainly comprises photosynthetic electron transport chain transcripts and other components that respond to light stimuli. According to taxonomic annotation, the majority of the genes represented in this hub come from green organisms. Transcripts involved in protein kinase activity were found in Hub 4 (23 transcripts), defined as “cyclin-dependent protein serine/threonine kinase regulator activity”. Hub 5 (25 transcripts) corresponded mainly to processes involved in genetic information processing, such as spliceosome, exosome, chromosome-associated proteins, or chaperones. Hub 6 (79 transcripts) is defined by several categories related to mitochondrial protein complexes and mitochondria transport, and has a central position in the network (connections to hubs 1, 2, 3 and 8). Hub 7 (46 transcripts) was defined by “DNA integrity checkpoint” ontology terms and consisted of cell cycle processes, such as transition from G1 phase to S or the previously mentioned DNA integrity checkpoint. Hub 8 (53 transcripts) was categorized as “response to temperature stimulus” and was composed mainly of transcripts that encode heat shock proteins. Components of hub 9 (22 transcripts) were related to “negative regulation of translation”. Overall, our 2500 most relevant genes appear to be distributed around the central role of the mitochondrion, whose origin traces back to the euglenozoan host cell [31]. In this respect, our taxonomic analysis specifically revealed that more than 10% of genes are related to kinetoplastids (the closest available proxy for the host cell) in all hubs, except for hub 3, categorized as “thylakoid” (Appendix A).

### 3.5. Cluster Annotation Enrichment Analysis and Gene Co-Expression

From the same top 2500 variable genes, we identified positive and negative relationships between pairs of genes based on gene expression. We tried to capture genes that behave conjointly across the various experimental conditions and group them into clusters. According to our expectations where a gene would be binary regulated (up or down), the optimal k solution should range between 2^5^ (32) and 2^13^ (8192) (accounting for 5 to 13 distinct experimental conditions with a total sample number of 23; see Table 1). We computed the optimal number of clusters and determined that 36 clusters was the most suitable solution for the selected genes (Appendix A). To better understand the underlying biological processes inside the clusters, ontologies that were overrepresented were extracted and analysed. Only five out of the 36 clusters were characterized by significantly overrepresented ontological terms (Appendix A). In total, those five clusters were composed of 631 transcripts out of the 2500 initially used for clustering, and 52% of them had at least one annotation attribute. Their expression can be visualized in hierarchically clustered heat maps (Figure 5).

Results from the enrichment tests revealed that “nucleosome category” was overrepresented in cluster 1, which contains transcripts of the “DNA damage checkpoint” and “ribosome” hubs of the ontological network, hub 1 and 2, respectively (see above). These transcripts encode histones, and core components of “nucleosome”, that participate in wrapping and compacting DNA into chromatin. The observation that DNA packaging, transcription and translation shared the same gene expression pattern may be relevant because in euglenids, as well as in dinoflagellates, chromosomes are permanently condensed [128]. Furthermore, transcripts encoding different components of the chloroplast reaction centres of hub 3 were also found in this cluster. This cluster was characterized by a larger down-regulated expression in PRJEB38787 (E), while other experiments were slightly over and under zero. Cluster 4 was enriched in “photosynthetic electron transport” and “DNA damage checkpoint” related terms mainly present in hub 3, with several transcripts encoding ATP synthase subunits in the former and cell cycle and apoptosis regulator proteins in the latter. Gene expression in cluster 4 was homogeneous with values ranging between one or minus one, except for a group of genes greatly down-regulated in studies PRJNA310762 (A), PRJEB10085 (B), PRJNA298469 (C), and likely to be not expressed in such experiments. About a third of the transcripts from cluster 19 encode different types of serine/threonine proteins and are ontologically typified by “cyclin-dependent protein serine/threonine kinase regulator activity”, which are processes closely related to cell cycle regulation. Their expression was slightly negative in the experiment PRJEB38787 (E) and positive in PRJEB10085 (B) while it remained unaltered in the rest of the experiments. “Neuroblast proliferation” and “neuroblast division” categories illustrated cluster 24, which, considering the unicellular nature of *E. gracilis*, was more likely to be related to cytoskeletal structure of eukaryotic cells formed during cell division or cell polarity than regulation of neurogenesis. In study PRJNA289402 (D), ABC transporters, fatty acid and polyketide synthesis were more down-regulated than in the remaining studies. Lastly, cluster 25 was enriched in “positive regulation of mitochondria organization” due to the presence of putative mitochondrial heat shock proteins that were co-regulated across studies. Besides, expression of cluster 25 was disparate for PRJNA289402 (D), compared with the other studies. A main difference was a group of transcripts largely downregulated in the PRJNA289402 (D) experiment, while they were upregulated in the remaining studies. Those transcripts putatively encode different components of the nitrogen metabolism, some chloroplastic electron transport chain components and ATP-dependent RNA helicase. A few transcripts related to cell cycle and translation, present in the annotation network, were found in cluster 25.

The cluster patterns reported above show that expression is driven by study rather than experimental conditions of the studies. Even if disappointing, these findings were similar after the tentative SVA correction of the batch effect present in the studies (Appendix A). Presumably, our approach was not able to properly capture the batch effect, maybe due to an unbalanced batch-group design of the studies [129]. Nonetheless, we observed that a selection of 133 genes, coding for the components of the photosynthetic and respiratory electron transport chains, were grouped together. This subset of genes, located in the chloroplast and in the mitochondrion, respectively, was selected because most of the experimental conditions (light/dark, presence or absence of acetate in the medium, oxic/anoxic environment) of the studies were expected to affect respiration and photosynthesis. As illustrated in Figure 6, the expression of these genes is also driven by the study rather than by the reported physico-chemical parameters of each experiment. Yet, most components of the mitochondrial electron transport chain among the 133 selected genes were grouped together after hierarchical clustering of their expression, while chloroplastic components exploded into different subgroups. Concretely, genes coding for light-harvesting complexes grouped together distantly from other chloroplastic components. These transcripts are nuclear-encoded and showed a taxonomic affinity to Streptophyta (Appendix A).

Overall, our last analysis indicates that genes that share common metabolic functions are packed together, as would be expected, even though the expression is driven by study rather than culture condition. Beyond the technical issues that may have contributed to a loss of exploitable signal (e.g., heterogeneous experimental “design”, see Table 1, uncorrected batch effects), these negative results can also be interpreted as additional evidence for the idea that, similar to what is known in trypanosomatids, nuclear gene expression in *E. gracilis* is not primarily regulated at the transcriptional level. In these parasites, gene regulation mostly occurs at the post-transcriptional level, through stabilization/degradation of mRNA molecules and control of mRNA translation (see [8] for a recent review of the issue). While the former mechanism should in principle change transcript abundance, the latter one might not be visible in comparative transcriptomics. For example, Yoshida et al. (2016) observed little change at the transcriptomic level following anaerobic treatment. Moreover, these changes in gene expression were inconsistent with respect to the activation of paramylon degradation and wax ester production [53]. In a more systematic investigation, Ebenezer et al. (2019) reported a striking lack of correlation between transcriptomic and proteomic data when comparing light and dark conditions [7]. As already mentioned, the raw transcriptomic data from these two studies were included in the present work (along with those of O’Neill et al. (2015) [52] and our own data), which allowed us to compare gene expression across a wider range of culture conditions at once. A few meaningful clusters of genes (i.e., following functional term enrichment) could be identified based on shared expression patterns across samples, which suggests that there is some biological signal in transcript abundance. However, the dominance of batch effects on these levels further questions the usefulness of transcriptomics for functional studies in *E. gracilis*.

## 4. Conclusions

Owing to its singular evolutionary origin, a merger between a chlorophyte alga and a phagotrophic unicellular belonging to a non-model eukaryotic group [20], *E. gracilis* is a fascinating, multifaceted chimeric organism, whose significance is constantly growing in domains as varied as the production of bio-based products [43], the treatment of wastewater ([130]), the provision of food supplements for space exploration [131], or the elucidation of mechanisms it shares with its parasitic trypanosome cousins [8,9,15] (see also the other articles of the present Special Issue).

By building a consolidated transcriptome of this photosynthetic eukaryote, we aimed at providing a solid resource to the community, taking into account previous work [7,52,53], yet enriched with unreleased data (obtained back in 2012–2014; Appendix A) [132]. Our final consensus transcriptome comprises 91,040 unique transcripts and 49,922 predicted non-redundant protein-encoding genes. It appears to be the most complete up-to-date, at least according to sequence metrics, the number of universal orthologs found, read percentages supporting the assembly, and the fact that most of the *E. gracilis* sequences available to date have been included. Hence, we have been able to capture more than 98% of the sequences produced in the other transcriptomes hitherto published, while the number of predicted genes is in the same range [7,53]. This suggests that there was still some room for improvement, contrary to expectations for the opposite [7], and it might be related to the inclusion of reads obtained without poly-A selection, but following DSN normalization.

Annotating these transcripts, whether from a functional or taxonomic point of view, remains a challenge, notably because of the lack of well-characterized closely related organisms, the trypanosomes being relatively derived parasites [133]. This results in a mere 26–27% of our predicted genes annotated by sequence similarity, above the 23% of Yoshida et al. (2016) [53], but below the 45% of O’Neill et al. 2015 [52] and the 52–55% of Ebenezer et al. (2019) [7], who further considered orthogroup sharing as annotation. In principle, this should encourage more large-scale studies, e.g., comparative transcriptomics performed in a wide range of culture conditions and stresses, in order to build a reliable gene expression network from co-expression data, and thereby provide alternative means for annotating genes of unknown function. Alas, as it now appears quite clearly, gene expression is mostly controlled at the post-transcriptional level in euglenozoans [7,8], including the regulation of chloroplast development in photosynthetic euglenids [134]. This implies that functional studies in *E. gracilis* have to be carried out through proteomics rather than transcriptomic approaches (e.g., [119,135]). This is fully possible considering the availability of several high-quality transcriptome assemblies to feed reference databases for proteomic fragment identification, including the one presented in this work. In this respect, the unfortunate lack of a complete genome beyond the draft level, even if frustrating, is not an insuperable issue [7].

Regarding the highly mixed taxonomic affinities of *Euglena* transcripts, our similarity searches yielded proportions in line with previous studies, even when those studies were based on more reliable phylogenetic approaches [136], such as the comprehensive work of Ebenezer et al. (2019) [7]. Altogether, the current knowledge points to the “shopping bag” [23,24,25] (or “red-carpet” [26]) model for the evolutionary origin of *Euglena*, i.e., transient endosymbioses during which multiple rounds of HGT/EGT have progressively shaped the plastid proteome. Yet, it is noteworthy that such a gene mixture would also be compatible with a kleptoplastidic origin for photosynthetic euglenids, in which the transient “endosymbioses” would actually imply stolen plastids and not intact symbionts. Moreover, some predatory euglenids, such as *Peranema trichophorum*, can feed either by phagocytosis of whole cells or by drilling a hole in their prey and then sucking up its cellular contents [137], a process known as myzocytosis [138]. Beyond providing a selective force for transferring genes to the host nucleus to service the ingested plastids, as in the recently characterized ARS (Antarctic Ross Sea) dinoflagellate bearing haptophyte-derived kleptoplastids [139], a kleptoplastidic model would also better fit the three membranes of the euglenid chloroplasts [20,140] and the presence of kleptoplastids acquired by myzocytosis in the early branching *Rapaza viridis* [141].

## Figures and Tables

**Figure 1 genes-12-00842-f001:**
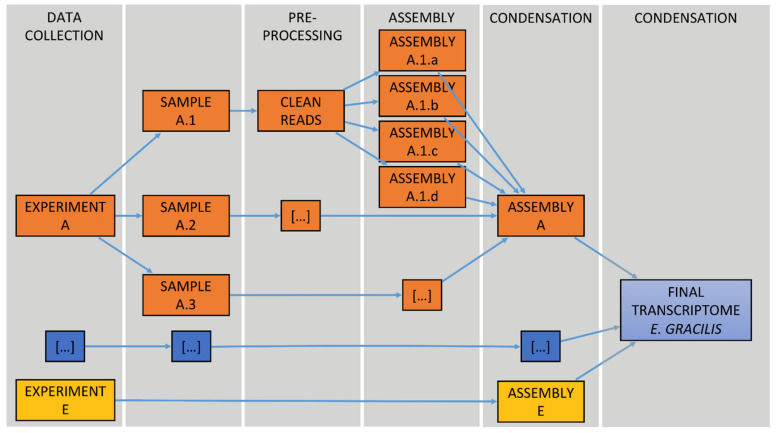
Schematic representation of our de novo transcriptome meta-assembly pipeline.

**Figure 2 genes-12-00842-f002:**
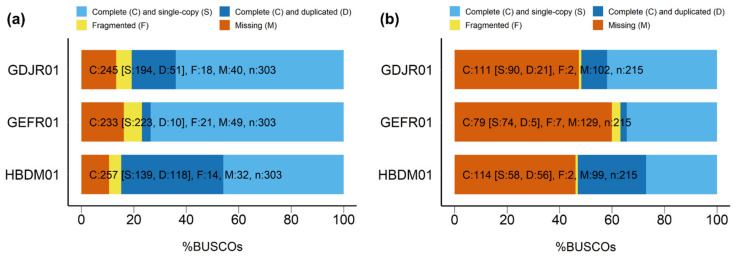
BUSCO-generated charts showing the relative completeness of the three public transcriptome assemblies, GEFR01 [7], GDJR01 [53] and HBDM01 (this study). BUSCO datasets were based on odb9. (**a**) “Eukaryota” (303 BUSCOs); (**b**) “Protists *ensembl*” (215 BUSCOs).

**Figure 3 genes-12-00842-f003:**
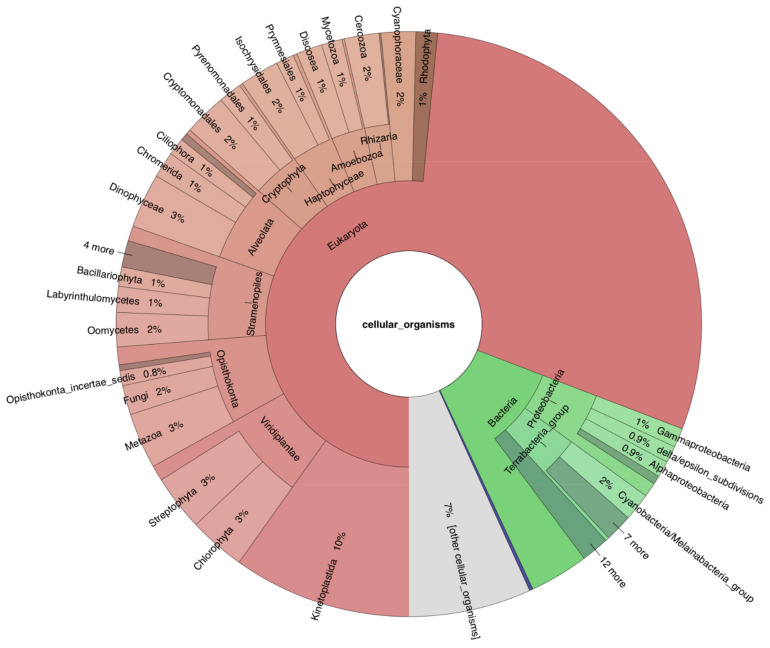
Taxonomic analysis of reconstructed transcripts (BLASTX MEGAN-like affiliations). The Krona chart is a zoom on the 13,850 transcripts to which a taxonomy could be associated, i.e., 28% of the 49,922 reconstructed transcripts. Among this classified fraction, 937 (7%) correspond to organisms to which we cannot assign a specific taxon (“other cellular organisms”). The thin blue slice is labelled “Archaea” (0.2%). The interactive chart is available as HTML Appendix A.

**Figure 4 genes-12-00842-f004:**
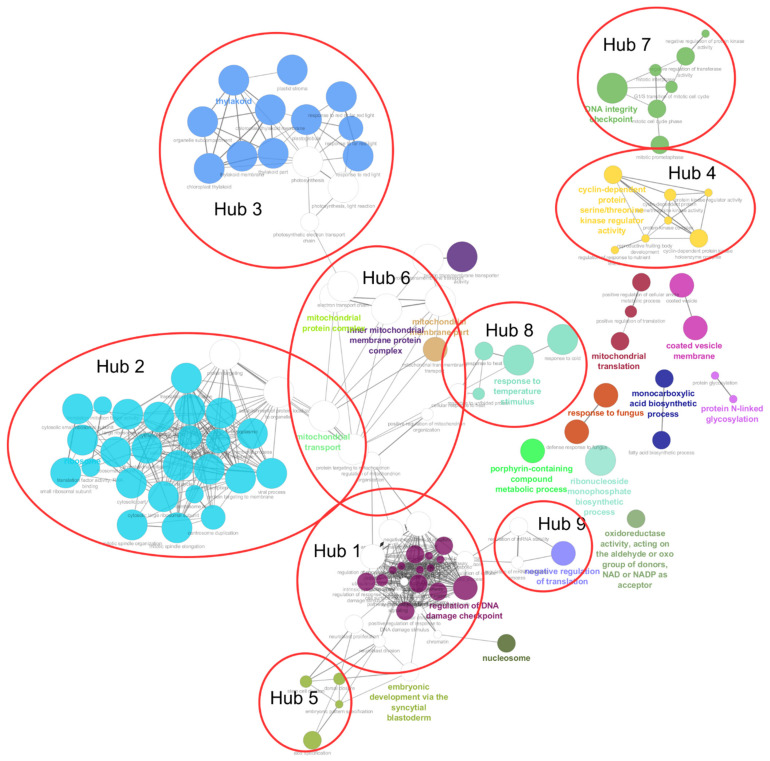
Annotation network of ontological terms showing the functional organization and relationships between the 2500 most variable genes. GO and KEGG terms were considered as a large pool in which the genes could be associated with 0 to N terms. Such associations served as the basis to infer the network (see text). Colours correspond to ontological terms (or groups of related ontological terms).

**Figure 5 genes-12-00842-f005:**
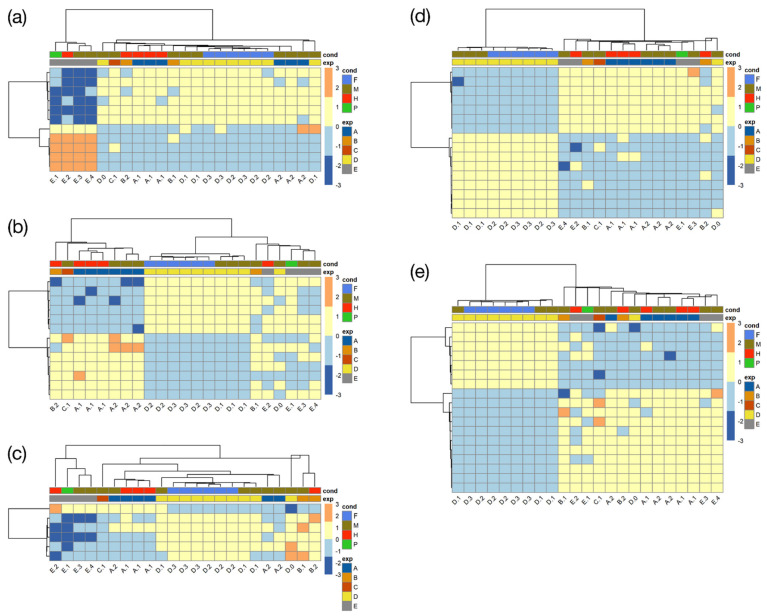
Selected co-expression clusters computed on the 2500 most variable genes. Only the five clusters characterized by significantly overrepresented ontological terms (featuring 631 transcripts) are shown. Heat maps and trees regroup samples behaving similarly across genes on the horizontal axis and genes behaving similarly across samples on the vertical axis; gene expression is vertically clustered to facilitate visualization (see text). Samples are colour-coded both by condition (F = fermentative, M = mixotrophic, H = heterotrophic, P = phototrophic) and by study (A = PRJNA310762, B = PRJEB10085, C = PRJNA298469, D = PRJNA289402, E = PRJEB38787). (**a**) Cluster 1; (**b**) cluster 4; (**c**) cluster 19; (**d**) cluster 24; (**e**) cluster 25.

**Figure 6 genes-12-00842-f006:**
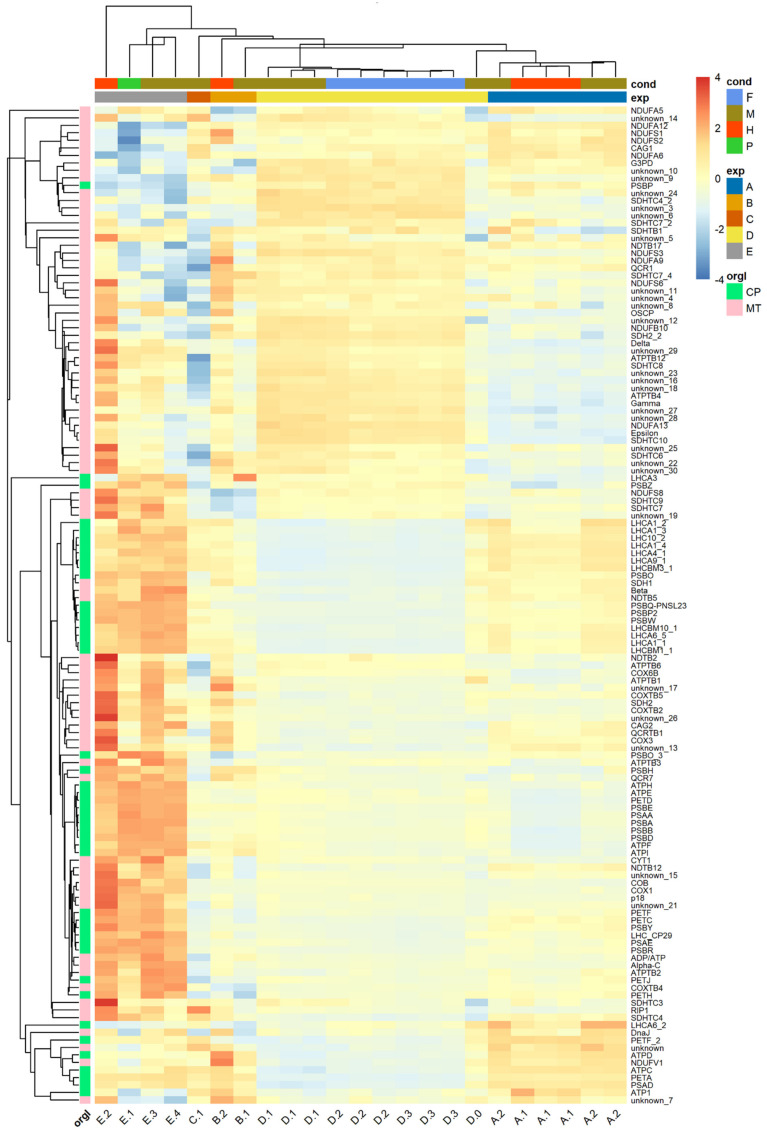
Expression heat map of 133 genes involved in electron transport chains. Heat maps and trees regroup samples behaving similarly across genes on the horizontal axis and genes behaving similarly across samples on the vertical axis (see text). Samples are colour-coded both by condition (F = fermentative, M = mixotrophic, H = heterotrophic, P = phototrophic) and by study (A = PRJNA310762, B = PRJEB10085, C = PRJNA298469, D = PRJNA289402, E = PRJEB38787). Genes are colour-coded by organelle (CP = chloroplast; MT = mitochondrion).

**Table 1 genes-12-00842-t001:** Representation of the collected data and overview of the experimental design. Exp. Code: letter assigned to each experiment (one letter per study). Study Acc.: public accession number of the BioProject. Sample Code: first letter corresponds to the experiment, first digit to experimental conditions of the samples, and second digit (if any) to the replicates. Run Acc.: public accession number of read FASTQ files. Temp.: estimated Celsius degrees of cell culture temperature. Medium: type of cell culture medium, rich (R) or mineral (M) plus carbon source (+C). Light: estimated light experimental conditions, darkness (D), low-light (LL) and high-light (HL). Shaking: rpm of shaker incubator. Cult. Cond.: trophic regime, fermentative (F), heterotrophic (H), phototrophic (P) or mixotrophic (M). Harvest Phase: development stage of the culture when collected, exponential phase (Exp) or stationary phase (Stat).

Exp. Code	Study Acc.	Sample Code	Run Acc.	Temp.	Medium	Light	Shaking	Cult Cond.	Harvest Phase	Reference
A	PRJNA310762	A.1.1	SRR3159774	25	R + C	D	0	H	Exp	[7]
		A.1.2	SRR3159775	25	R + C	D	0	H	Exp	
		A.1.3	SRR3159776	25	R + C	D	0	H	Exp	
		A.2.1	SRR3159777	25	R + C	LL	0	M	Exp	
		A.2.2	SRR3159778	25	R + C	LL	0	M	Exp	
		A.2.3	SRR3159779	25	R + C	LL	0	M	Exp	
B	PRJEB10085	B.1	ERR974915	21	M + C	LL	0	M	Stat	[52]
		B.2	ERR974916	30	R+C	D	200	H	Stat	
C	PRJNA298469	C.0	SRR2628535	25	M	LL	0	M	Stat	[7]
D	PRJNA289402	D.0	SRR3195326	26	R+C	HL	120	M	Stat	[53]
		D.1.1	SRR3195327	26	R+C	HL	120	M	Stat	
		D.1.2	SRR3195329	26	R+C	HL	120	M	Stat	
		D.1.3	SRR3195331	26	R+C	HL	120	M	Stat	
		D.2.1	SRR3195332	26	R+C	HL	120	F	Stat	
		D.2.2	SRR3195334	26	R+C	HL	120	F	Stat	
		D.2.3	SRR3195335	26	R+C	HL	120	F	Stat	
		D.3.1	SRR3195338	26	R+C	HL	120	F	Stat	
		D.3.2	SRR3195339	26	R+C	HL	120	F	Stat	
		D.3.3	SRR3195340	26	R+C	HL	120	F	Stat	
E	PRJEB38787	E.1	ERR4227585	25	M	LL	100	P	Exp	This study
		E.2	ERR4227586	25	M+C	D	100	H	Exp	
		E.3	ERR4227587	25	M+C	LL	100	M	Exp	
		E.4	ERR4227588	25	M+C	HL	100	M	Exp	

**Table 2 genes-12-00842-t002:** Basic statistics based on transcript properties of reconstructed transcriptomes from collected data. ACC: study accession, REF: bibliographic reference, RAW: number of downloaded reads, PRE: number of good reads after pre-processing, CNT: number of reads removed after pre-processing considered as contamination (reads per million; rpm), SEQ: number of transcripts, MIN: minimal sequence length, MAX: maximal sequence length, MEAN: mean sequence length, TOTAL: combined sequence length, SEQ < 200: number of transcripts under 200 n, SEQ > 1 k: number of transcripts over 1000 nt, SEQ > 10 k: number of transcripts over 10,000 nt, ORF: number of sequences with a predicted open reading frame, ORF (%): for contigs with an ORF, the mean % of the contig covered by the ORF, N[z]: minimum contig length needed to cover [z]% of the transcriptome. GC (%): percentage of guanine-cytosine content, PART and PART (%): number and percentage of sequences contributed to the final consensus transcriptome (see below). In PRJEB10085 (B) (filtered), sequences <500 nt were further discarded (see text).

Statistic	A	B	B (Filtered)	C	D	E
ACC	PRJNA310762	PRJEB10085	PRJEB10085	PRJNA298469	PRJNA289402	PRJEB38787
REF	[7,52,53]					This study
RAW	61,531,862	383,416,636	383,416,636	27,096,926	285,148,782	1,902,226,200
PRE	57,862,467	310,302,570	310,302,570	25,244,887	267,779,751	875,299,135
CNT	740 (12 rpm)	9080 (29 rpm)	9080 (29 rpm)	1750 (68 rpm)	1191 (4 rpm)	2403 (2 rpm)
SEQ	38,559	95,490	36,287	42,363	37,425	51,021
MIN	101	101	500	101	101	101
MAX	13,929	21,744	21,744	11,354	26,839	10,795
MEAN	1043	647	1312	810	1120	610
TOTAL	40,861,413	64,426,688	47,615,807	34,438,742	42,382,170	31,671,589
SEQ < 200	4330	17,074	0	782	3051	3989
SEQ > 1 k	16,289	18,638	18,638	10,932	17,048	7104
SEQ > 10 k	4	15	15	1	13	1
ORF	24,757	29,060	27,842	27,063	24,817	26,882
ORF (%)	88%	82%	83%	89%	87%	93%
N90	576	347	654	419	606	367
N70	1140	667	1101	686	1187	528
N50	1607	1282	1574	1014	1658	753
N30	2257	2033	2243	1452	2318	1090
N10	3600	4026	3707	2358	3812	1850
GC (%)	64%	58%	62%	64%	64%	64%
PART	22,234	-	27,730	10,129	19,663	11,602
PART (%)	24.3%	-	30.3%	11.1%	21.5%	12.7%

**Table 3 genes-12-00842-t003:** Basic statistics of transcript properties computed for the three public transcriptome assemblies, including the consensus transcriptome generated in the present work, and completed with data retrieved from the publications of Ebenezer et al. (2019) [7] and Yoshida et al. (2016) [53]. Row titles are as in Table 2, except for CDS: number of unique coding sequences (i.e., ORFs or UNIGENEs), GMS-T and GMS-T (%): number and percentage of predicted protein coding regions calculated by GeneMarkS-T.

Statistic	GEFR01	GDJR01	HBDM01
REF	[7,53]		This study
SEQ	72,506	113,152	91,040 ^1^
MIN	202	201	201
MAX	25,763	21,553	26,839
MEAN	869	1087	1096
TOTAL	63,049,595	122,976,775	100,187,451
SEQ < 200	0 ^1^	0 ^1^	0 ^1^
SEQ > 1 k	19,740	49,277	37,294
SEQ > 10 k	25	27	24
ORF ^2^	30,467	65,943	62,287
ORF (%)	79%	73%	85%
N90	374	523	545
N70	704	1130	965
N50	1242	1604	1432
N30	1916	2181	2049
N10	3344	3347	3410
GC (%)	61%	63%	63%
CDS	32,128	49,826	49,922
GMS-T	35,929	63,432	58,542
GMS-T (%)	49%	56%	64%

^1^ Submission tools for sequence repositories do not accept transcripts ≤ 200 nt. Hence, the number of sequences in the public version of HBDM01 is lower than reported elsewhere in this work. ^2^ ORFs were determined with TransDecoder, whereas CDS were determined with EvidentialGene (or a similar tool, depending on the study).

**Table 4 genes-12-00842-t004:** Mapping fraction of pre-processed reads from each collected dataset (rows) to the three public transcriptome assemblies (columns), GEFR01 [7], GDJR01 [53] and HBDM01 (this study).

Code	Accession	Reference	GEFR01	GDJR01	HBDM01
A	PRJNA310762	[7]	87.40%	92.51%	93.38%
B	PRJEB10085	[52]	84.68%	90.13%	91.49%
C	PRJNA298469	[7]	80.26%	91.66%	90.39%
D	PRJNA289402	[53]	85.25%	95.04%	94.28%
E	PRJEB38787	This study	75.28%	51.39%	80.76%

**Table 5 genes-12-00842-t005:** TransRate and Detonate assembly scores for the three public transcriptome assemblies, GEFR01 [7], GDJR01 [53] and HBDM01 (this study). Scores indicate how well transcripts are supported by the RNA-Seq data.

Assembly Score	GEFR01	GDJR01	HBDM01
TransRate Score	0.1789	0.0304	0.0430
TransRate Optimal Score	0.2051	0.1729	0.0764
Detonate Score	−97,461 × 10^6^	−97,561 × 10^6^	−97,459 × 10^6^

**Table 6 genes-12-00842-t006:** SL-sequence related statistics for the three public transcriptome assemblies, GEFR01 [7], GDJR01 [53] and HBDM01 (this study). These correspond to exact matches limited to the first 40 nucleotides of each transcript.

Threshold (nt)	Statistic	GEFR01	GDJR01	HBDM01
24	Forward matches	24	5	86
	Reverse matches	21	0	114
	Total matches	45	5	200
	Average length (nt)	24.00	24.00	24.00
14	Forward matches	176	16,580	3370
	Reverse matches	200	12,999	3265
	Total matches	376	29,579	6635
	Average length (nt)	16.28	15.57	15.59
12	Forward matches	749	18,322	4403
	Reverse matches	766	16,016	5397
	Total matches	1515	34,338	9800
	Average length (nt)	13.37	15.19	14.68

## Data Availability

Publicly available datasets were generated in this study. These can be found at the European Nucleotide Archive (ENA), under study accession PRJEB38787 and TSA project accession HBDM01000000. Most custom analysis scripts have been deposited on GitHub (https://github.com/microalgues/clustering (accessed on 22 August 2019)).

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
