# Peer review of "De Novo Transcriptome Meta-Assembly of the Mixotrophic Freshwater Microalga Euglena gracilis"

_genes, 2021, doi:10.3390/genes12060842_

Round 1
Reviewer 1 Report
Cordoba et al present a study focusing on the construction of an enhanced assembly of the total transcriptome of the photosynthetic alga Euglena gracilis.
E. gracilis is one of the oldest and most studied algal model organisms, mostly because it is easy to manipulate and can grow in many conditions ranging from phototrophy to mixotrophy or heterotrophy. This enables the molecular and genetic study of many metabolic pathways that can be triggered via culture conditions. However, the complexity of the genetic structure of Euglena, both structurally and evolutionary, has greatly impaired its use as a forward or reverse genetic model. The estimated large size large and heterogeneity of the nuclear genome of Euglena makes it difficult to sequence and assemble. For instance, a draft assembly published in 2019 combining several technologies of DNA reads could only cover an estimated 20% of the gene content of the nuclear genome. Therefore, the sequencing of total transcripts has been prefered, first using EST and later using whole transcriptome NGS sequencing. This paper aims at providing the most complete and finest transcriptome repertoire to date by blending new data/assembly together with previous ones.
This manuscript is comprehensive but also very technical, which makes the reading rather arduous. The authors first present previous data releases with emphasis on quality and size. This serves as a motivation for including or discarding them in their own study. They then describe the process of individual assembly (or reassembly), decontamination, annotation, and merging of all read datasets collected from the literature or produced by them. This is followed by a description of the blending and dereplication of all transcript collections into a consensus transcriptome. The functional repertoire of this assemblage is assessed and genes are assigned to a taxonomic “affinity” by similarity searches. Finally, a tentative differential expression clustering and network analysis is proposed, which spans over all available sample and culture conditions, past and new.
- Assembly
5 merged assemblies were produced for this manuscript by combining individual sample assemblies (one per original data source, 4 existing, one new). Those are compared with each other in terms of size and quality. It appears that the quality of the reads specifically produced for this study is surprisingly low: 56% of them were discarded after quality control, while less than 10% were discarded from other datasets. The authors do not really discuss this point and the possible consequences on their own assembly. For instance, the author’s merged assembly is globally smaller and of “lower quality” than the one of Yoshida et al, although they had potentially much more reads/coverage available. The number of contigs recovered is larger, but their mean length is much shorter compared to Yoshida et al, which is a potential sign of higher fragmentation of transcripts.
The final consensus transcriptome was produced by combining all transcript sequences from the 5 merged assemblies into a single collection. Redundancy was first reduced by selecting the best sequence per alternate transcript variants. Then potential isoforms were reduced to keep only one representative CDS per homologous transcripts/protein. This blended transcriptome is compared to those previously published using multiple tools and metrics. The new consensus transcriptome is globally comparable to previous ones, with no real change in the number of transcripts and of non-redundant coding sequences compared to the latest publication (GDJR01). If adding sequencing effort does not change transcript count, this may indicate that most of the genes encoded in the genome of Euglena have been captured.
Additionally, the original high quality reads used to produce the 5 merged assemblies were mapped to determine the respective coverage of each collection against 3 consensus transcriptome sequences. This is summarized in table 4, which, in my opinion, is intriguing. Reads from the present study map poorly against former consensus sequence collections. I believe this can have two meanings : 1) Those new reads contain a lot of errors (they seem of no great quality after all) and are rejected by the mapping software. 2) Those new reads are correct but were generated from a slightly different biological entity (I’ll explain my point below). Table 4 also shows that all old read sets have high mapping counts against any consensus transcriptome, including the new one. Considering that the number of transcripts was not drastically changed in this new consensus, this means that more sequences from previous assemblies were picked as “best transcript” by EvidentialGene, and that, logically, reads generated in the present study fail to map on them, explaining the lower mapping score. In conclusion, the actual enhancement provided by the present sequencing effort was limited. This is also visible in the BUSCO analysis in Fig 2, where it appears that the duplication level of transcripts corresponding to core genes is higher in the new consensus. This may have at least two origins in my opinion : 1) New data from this study introduced more fragmented transcripts in the consensus, and fragments of the same core gene are counted twice. 2) Two isoforms of the same core genes were actually sequenced and merged in the consensus transcriptome (and eventually dereplicated afterwards into non-redundant ORFs). This latter points (and point 2 above) leads me to a fundamental question (potentially stupid, please tell me) : are low mapping and busco scores only due to bad quality of the new reads? or is there a chance that the strains used for all those studies (past and present) have genetic differences? Are there additional analyses (like phylogenies of similar transcripts) already made by the authors that could enlighten this issue?
I also have an additional technical question : Did the authors try to do co-assemblies, either study centered, or even all together? Is there a fundamental problem against doing it?
- Functional and taxonomic analysis
One of the strengths of this study is to assemble reads generated from cells cultivated in 4 different conditions to try to enlarge the expressed gene repertoire (and thus recruit more transcripts in the final assembly compared to what was already observed); but also to try to detect differential expression patterns. It seems, from assembly statistics, that the first goal was not successful, which gives an important information about the state of coverage of the transcriptome of Euglena. Concerning the later objective, the results presented in the manuscript are contrasted. The signal of differential expression between genes was sufficiently powerful to reconstruct a network of co-expression where sub-networks correspond to metabolic modules. This implies that related functions are expressed and/or regulated somehow conjointly in the cell. On the other hand, the analysis of differential expression of genes between samples is rather disappointing. The authors compared the expression of around 2000 of the most expressed genes across all sub-samples using every data set included in the manuscript. The expectation was that samples grown in similar conditions would look more alike. Unfortunately, results show that sub-samples cluster by study, meaning that experiment driven bias bears the strongest statistical signal, forbidding any conclusion about the regulation of expression in relation to physiology. The authors consider that their cross-sample results support the idea that regulation in Euglena does not happen at transcription level. My opinion is that those results don’t have the power to either support or refute this hypothesis, because of the strong technical bias. Conversely, the fact that across genes expression data has sufficient signal to cluster genes into metabolically related networks is very interesting and is compatible with metabolism co-regulation via transcription.
Finally, the taxonomic affiliation of ORFs led to results in accordance with previously published papers on the subject: The genome of Euglena is a complex patchwork of genes of multiple origins that accumulated over time via endosymbioses and horizontal gene transfers.
- General observations.
Enhancing the sequencing and annotation of important biological models is a crucial task that should be encouraged. Unfortunately, the main risk is to end up with moderately new results given the effort. Those should anyway be rewarded because of the service offered to the community, and because even disappointing results are useful. Thus, I believe that this manuscript should be published, but requires some modifications. I think that more introspective discussion about the quality of the data produced is necessary, and I really would like the authors to elaborate more about this surprising low mapping of reads against old and new consensus transcriptomes. I also think they should reconsider their conclusion about the absence of transcription based regulation. Finally, I believe that the quality of the writing and the use of english are good, even if the style is very technical.
Author Response
Cordoba et al present a study focusing on the construction of an enhanced assembly of the total transcriptome of the photosynthetic alga Euglena gracilis.
E. gracilis is one of the oldest and most studied algal model organisms, mostly because it is easy to manipulate and can grow in many conditions ranging from phototrophy to mixotrophy or heterotrophy. This enables the molecular and genetic study of many metabolic pathways that can be triggered via culture conditions. However, the complexity of the genetic structure of Euglena, both structurally and evolutionary, has greatly impaired its use as a forward or reverse genetic model. The estimated large size large and heterogeneity of the nuclear genome of Euglena makes it difficult to sequence and assemble. For instance, a draft assembly published in 2019 combining several technologies of DNA reads could only cover an estimated 20% of the gene content of the nuclear genome. Therefore, the sequencing of total transcripts has been prefered, first using EST and later using whole transcriptome NGS sequencing. This paper aims at providing the most complete and finest transcriptome repertoire to date by blending new data/assembly together with previous ones.
>>> Thank you for this accurate summary of our work.
This manuscript is comprehensive but also very technical, which makes the reading rather arduous. The authors first present previous data releases with emphasis on quality and size. This serves as a motivation for including or discarding them in their own study. They then describe the process of individual assembly (or reassembly), decontamination, annotation, and merging of all read datasets collected from the literature or produced by them. This is followed by a description of the blending and dereplication of all transcript collections into a consensus transcriptome. The functional repertoire of this assemblage is assessed and genes are assigned to a taxonomic “affinity” by similarity searches. Finally, a tentative differential expression clustering and network analysis is proposed, which spans over all available sample and culture conditions, past and new.
>>> Again. These were indeed our objectives for this study. We apologize for the very technical writing, which was selected because our focus was on the methods for generating such a meta-assembly rather than on the biological outcome of the analyses. Our idea was that the approaches we used in this manuscript would be useful for colleagues working on other protists’ transcriptomes, hence the sometimes excessive amount of details or rationale (see comments of reviewer 2). Besides, we have other papers under review for the biological aspects of our work (e.g., Gain et al. 2021, currently in the second round of review for New Phytologist). Anyway, for the sake of clarity and brevity, we amended our text to streamline it as much as possible, notably by deleting apparent repetitions between Methods and Results and Discussion.
Assembly
5 merged assemblies were produced for this manuscript by combining individual sample assemblies (one per original data source, 4 existing, one new). Those are compared with each other in terms of size and quality. It appears that the quality of the reads specifically produced for this study is surprisingly low: 56% of them were discarded after quality control, while less than 10% were discarded from other datasets. The authors do not really discuss this point and the possible consequences on their own assembly. For instance, the author’s merged assembly is globally smaller and of “lower quality” than the one of Yoshida et al, although they had potentially much more reads/coverage available. The number of contigs recovered is larger, but their mean length is much shorter compared to Yoshida et al, which is a potential sign of higher fragmentation of transcripts.
>>> First, apparently, we did not make sufficiently clear that our sequencing reads had been obtained in a different way than in the other studies. Indeed, we did not fish out mRNAs using their poly-A tail, but used a RNA-seq protocol based on DSN normalization (with a thermostable Duplex-Specific Nuclease) to reduce the amount of rRNA molecules in an attempt to sequence mRNAs that might have lacked a poly-A tail. By reducing the proportion of reads that come from highly expressed RNAs, DSN also increases the reads that come from rarer transcripts without requiring deeper sequencing. We have now better explained this point at the beginning of the Results and Discussion and in the Conclusions. From the manufacturer’s (Evrogen) leaflet: “DSN normalization is performed after RNA-seq sample preparation and before cluster generation. It involves the degradation of abundant DNA molecules derived from rRNA, tRNA, and housekeeping genes while preserving DNA molecules derived from less abundant transcripts. This method can be useful in a wide range of applications, including transcriptome discovery and annotation, the analysis of bacterial transcriptomes that lack poly-A tails, and the analysis of highly degraded RNA from sources such as FFPE.” [http://nextgen.mgh.harvard.edu/attachments/DSN_Normalization_SamplePrep_Guide_15014673_B.pdf]
>>> Second, reads from the other studies were downloaded from a public repository (NCBI SRA) and might have been enriched in reads actually retained to build the corresponding assemblies. In the case of our own reads, we started from all raw reads (and indeed, the reverse sequences were of lower quality than the forward sequences, which led us to filter out 50% of raw reads, see section 3.1.1). However, even if our raw reads represent 70% of the 2.6 billion reads available at the beginning of the work, our study (E) did not over-contribute (rather the opposite, as you acutely observe) to the final consensus transcriptome (see Table 2), with 12.7% of the transcripts tracing back to our reads (other studies ranging from 11.1% to 30.3%). In our opinion, this suggests that our methodology captured the necessary reads independently of the number of initial raw reads.
>>> Third, regarding fragmentation, we should distinguish two aspects. While our own individual assembly indeed shows shorter transcripts on average (610 nt, see Table 1) than experiments A (1043 nt) and D (1120 nt), it is less different from assemblies from experiments B (647 nt pre-filtration) and C (810 nt). However (as pointed out in your next comment), our contribution in this manuscript is the final consensus assembly (HBDM01, Table 3), in which the average transcript length is 1096 nt, to be contrasted with 869 nt for GEFR01 (Ebenezer et al. 2019) and 1087 nt for GDJR01 (Yoshida et al. 2016). Our final transcripts are thus not shorter. Moreover, the BUSCO analysis (Figure 2) shows that our assembly is the most complete and includes the smallest fraction of fragmented genes (yet at the expense of increased redundancy), whether evaluated on “Eukaryota” or “Protists ensembl” datasets (see also next comments).
The final consensus transcriptome was produced by combining all transcript sequences from the 5 merged assemblies into a single collection. Redundancy was first reduced by selecting the best sequence per alternate transcript variants. Then potential isoforms were reduced to keep only one representative CDS per homologous transcripts/protein. This blended transcriptome is compared to those previously published using multiple tools and metrics. The new consensus transcriptome is globally comparable to previous ones, with no real change in the number of transcripts and of non-redundant coding sequences compared to the latest publication (GDJR01). If adding sequencing effort does not change transcript count, this may indicate that most of the genes encoded in the genome of Euglena have been captured.
>>> We generally agree with this comment. To put this in perspective with your previous concerns, starting from much more reads and recovering a consensus transcriptome that is not that different from the assemblies that are already available, even though not very exciting, indicates that, as a community of researchers, we are approaching the biological reality of Euglena gracilis. It also suggests that our methods are robust.
Additionally, the original high quality reads used to produce the 5 merged assemblies were mapped to determine the respective coverage of each collection against 3 consensus transcriptome sequences. This is summarized in table 4, which, in my opinion, is intriguing. Reads from the present study map poorly against former consensus sequence collections. I believe this can have two meanings : 1) Those new reads contain a lot of errors (they seem of no great quality after all) and are rejected by the mapping software. 2) Those new reads are correct but were generated from a slightly different biological entity (I’ll explain my point below). Table 4 also shows that all old read sets have high mapping counts against any consensus transcriptome, including the new one. Considering that the number of transcripts was not drastically changed in this new consensus, this means that more sequences from previous assemblies were picked as “best transcript” by EvidentialGene, and that, logically, reads generated in the present study fail to map on them, explaining the lower mapping score. In conclusion, the actual enhancement provided by the present sequencing effort was limited. This is also visible in the BUSCO analysis in Fig 2, where it appears that the duplication level of transcripts corresponding to core genes is higher in the new consensus. This may have at least two origins in my opinion : 1) New data from this study introduced more fragmented transcripts in the consensus, and fragments of the same core gene are counted twice. 2) Two isoforms of the same core genes were actually sequenced and merged in the consensus transcriptome (and eventually dereplicated afterwards into non-redundant ORFs). This latter points (and point 2 above) leads me to a fundamental question (potentially stupid, please tell me) : are low mapping and busco scores only due to bad quality of the new reads? or is there a chance that the strains used for all those studies (past and present) have genetic differences? Are there additional analyses (like phylogenies of similar transcripts) already made by the authors that could enlighten this issue?
>>> Thank you for these insightful comments. We globally share your concerns but we are not convinced that digging into the issue will bring anything more than what we already have. Specifically, our reads may be less good than the other publicly released (see above) reads, but our reads were obtained differently (no poly-A selection but DSN normalization) and they are also much more numerous. Thus, we have enough “good” reads to not include “bad” reads in the final consensus transcriptome. Moreover, as you correctly point out, the latter is dereplicated afterwards into non-redundant ORFs. Nevertheless, we have added a short discussion about the relatively poor mapping of our reads (Table 4) in response to your comment and to a similar comment of reviewer 2: “It is probable that our reads have a lower mapping percentage because they were generated from DSN-normalized total RNA samples, for which analyses of a preliminary sequencing lane revealed many reads corresponding to non-mRNA sequences (e.g., rRNA). However, the specifically low mapping to GDJR01 (D) cannot be explained easily because “transcripts” matching to rRNA sequences were identified in all three public transcriptomes (Archive 1).”
>>> Regarding the strain issue, it is indeed intriguing. In principle, four studies, including our own, have sequenced the same strain, known as the “strain Z” (SAG 1224-5/25), at least based on BioSample metadata. However, O’Neill et al. (2015, PRJEB10085) mention that they sequenced a different strain: “Euglena gracilis var. saccharophila Klebs (SAG 1224/7a)”. Yet, when Ebenezer et al. (2019) recycles these reads along with newly generated reads, they say that they sequenced the “strain Z1 provided by William Martin”. Thus, the question is quite messy, but if a transcriptome has to be different from the others, it should rather be the original one described in O’Neill et al. (2015). We have added a sentence about this in the manuscript, when discussing an odd sub-population of transcripts in the corresponding experiment (see our answer to reviewer 2).
>>> In an attempt to address your point, we have performed an additional analysis based on a global clustering of all transcripts at the nucleotide level (see end of section 3.2.2). This work provided a new Table (S1b) and a new multipanel Figure (S4). Of course, it is possible to build trees from these clusters and we actually did it for a few hundreds of them. However, these trees only composed of Euglena gracilis sequences are not easy to summarize nor interpret. That is why we prefer not to mention these phylogenetic analyses in the revised manuscript, even though we could provide the trees if you deem them useful.
I also have an additional technical question : Did the authors try to do co-assemblies, either study centered, or even all together? Is there a fundamental problem against doing it?
>>> In spite of the quite good configuration of our grid computer (one node with 512 GB of RAM at the time), we did not manage to assemble all reads at once using Trinity. That is why we used the sequential approach outlined in Figure 1. Moreover, from a technical point of view, it was interesting to see how the individual datasets behaved before combining them and to estimate the respective contribution of each one to the final consensus transcriptome. Nonetheless, with hindsight, we acknowledge that the whole exercise might look overkill...
Functional and taxonomic analysis
One of the strengths of this study is to assemble reads generated from cells cultivated in 4 different conditions to try to enlarge the expressed gene repertoire (and thus recruit more transcripts in the final assembly compared to what was already observed); but also to try to detect differential expression patterns. It seems, from assembly statistics, that the first goal was not successful, which gives an important information about the state of coverage of the transcriptome of Euglena. Concerning the later objective, the results presented in the manuscript are contrasted. The signal of differential expression between genes was sufficiently powerful to reconstruct a network of co-expression where sub-networks correspond to metabolic modules. This implies that related functions are expressed and/or regulated somehow conjointly in the cell. On the other hand, the analysis of differential expression of genes between samples is rather disappointing. The authors compared the expression of around 2000 of the most expressed genes across all sub-samples using every data set included in the manuscript. The expectation was that samples grown in similar conditions would look more alike. Unfortunately, results show that sub-samples cluster by study, meaning that experiment driven bias bears the strongest statistical signal, forbidding any conclusion about the regulation of expression in relation to physiology. The authors consider that their cross-sample results support the idea that regulation in Euglena does not happen at transcription level. My opinion is that those results don’t have the power to either support or refute this hypothesis, because of the strong technical bias. Conversely, the fact that across genes expression data has sufficient signal to cluster genes into metabolically related networks is very interesting and is compatible with metabolism co-regulation via transcription.
>>> Again, thank you for this very accurate depiction of our work. The only detail that is wrong is that we focused on the 2500 genes showing the most variable expression, not the 2000 top-expressed genes. Regarding the interpretation for the lack of congruence in gene expression across similar conditions, we did try to correct for experimental bias (batch effect) using SVA (see Figure S8). However, it did not work at all, which suggests that the transcriptional regulation is not massively impacted by the contrasted conditions, such as high light vs. low light high vs. darkness or fermentation vs. heterotrophy vs. mixotrophy vs. autotrophy (see Table 1). This is also visible at a small scale, when focusing on 133 genes functioning in electron transfer chains (Figure 6). Moreover, in the aforementioned manuscript for New Phytol, we present proteomic abundances (iTRAQ data) that were generated in the same culture conditions as those in which we produced our RNA-seq reads. To address your concern, we compared the relative abundances of circa 120 transcripts and proteins in mixotrophic conditions. This value is negative for approximately one third of the comparisons under the two mixotrophic conditions, indicating that the relative variation in the abundance of these transcripts is of opposite sign to that of the corresponding protein (see attached Excel sheet). Finally, even if the two other reviewers are quite convinced by this lack of transcriptional regulation, we have added a sentence in the Conclusions in response to a comment of reviewer 3 that might also suit your opinion: “... nuclear gene expression in E. gracilis is not primarily regulated at the transcriptional level. In these parasites, gene regulation mostly occurs at the post-transcriptional level, through stabilization/degradation of mRNA molecules and control of mRNA translation [see Vesteg et al. 2019] for a recent review of the issue). **While the former mechanism should in principle change transcript abundance, the latter one might not be visible in comparative transcriptomics.**” Further, we have slightly changed a subsequent sentence: “A few meaningful clusters of genes (i.e., following functional term enrichment) could be identified based on shared expression patterns across samples, which suggests that there is some biological signal in **transcript abundance**.”
Finally, the taxonomic affiliation of ORFs led to results in accordance with previously published papers on the subject: The genome of Euglena is a complex patchwork of genes of multiple origins that accumulated over time via endosymbioses and horizontal gene transfers.
>>> Indeed. That is why we did not invest a lot of time in these analyses (but see our answers to the other reviewers’ comments).
General observations
Enhancing the sequencing and annotation of important biological models is a crucial task that should be encouraged. Unfortunately, the main risk is to end up with moderately new results given the effort. Those should anyway be rewarded because of the service offered to the community, and because even disappointing results are useful. Thus, I believe that this manuscript should be published, but requires some modifications. I think that more introspective discussion about the quality of the data produced is necessary, and I really would like the authors to elaborate more about this surprising low mapping of reads against old and new consensus transcriptomes. I also think they should reconsider their conclusion about the absence of transcription based regulation. Finally, I believe that the quality of the writing and the use of english are good, even if the style is very technical.
>>> We are very grateful for this last comment. We did our best to address your concerns. We hope that the revised manuscript will live up to your expectations.

Reviewer 2 Report
The authors present a meta-assembly based on five previously published transcriptomes and new sequencing data in order to somewhat remediate the absence of a satisfactory genomic dataset of E. gracilis by obtaining a transcriptome of so far the highest quality and degree of completeness. They subsequently use the dataset as reference to attempt a differential transcriptomic analysis using the reads generated in the previous studies and determine co-regulated gene clusters.
The attention dedicated to the various steps of the assembly process itself, such as multiple rounds of read-filtering, decontamination, pre-assemblies and general fine-tuning of the whole meta-assembly strategy (as far as I know not often employed in protist -omics) is extensive and impressive, so is the rigorous statistics behind the expression analysis; these are undoubtedly the main strength of this work.
Unfortunately, the biological (functional and evolutionary) conclusions that could be drawn from the results of the gene expression analysis are quite limited and relatively unremarkable. This is largely due to the fact, that the admittedly complex modulation of E. gracilis proteome in response to different conditions takes place mainly at translational, not transcriptional level. This has been strongly suspected for some time and the authors were aware of it. As much as I find it important (albeit emotionally unrewarding) to demonstrate this fact in such comprehensive and definitive way, I would expect the authors to provide stronger reasoning for dedicating time to what is essentially re-attempting (albeit more rigorously and in depth) something that was previously noted to be relatively unlikely to yield very novel and conclusive findings, for example by pointing out flaws of the previous studies, discussing the potential room for improvement or citing studies where such a strategy paid off.
While the inter-study gene expression analysis does not bring conclusive results, the network of co-regulated gene clusters connects at least some of the proteins into functionally distinct “hubs” and represents a useful resource at intra-study level and could be elaborated on more in the text, especially in regard to discussing the potential biological interpretation. Generally, the Discussion section of the paper is relatively descriptive and dry (a bit like if it were a continuation of Results) and would benefit from more attempts at interpretation of the data in relation to other, non-omics disciplines of Euglena studies (biochemistry, molecular biology, ecology etc.) and theorizing. I include suggestions/questions on the respective lines of the text, see below. For instance, I was very surprised by the profound difference in the proportion of trans-spliced transcripts identified in this study in comparison to previous works and frequently cited values and would be very interested to hear more of the authors opinion on what could be the story behind this phenomenon, how it could relate to what we know about euglenozoan trans-splicing, what could be the biological significance of it etc. Related to this, I also wonder what is the state of 5’-end completeness of the transcripts, especially compared to the previously published datasets.
Lastly, the authors also performed a transcriptome-wide screening of phylogenetic origin/affiliation of the E. gracilis proteins which is unfortunately based on best BLAST hit approach only which is relatively superficial (especially compared to the analogous screening performed in some previous studies that employed large scale tree generation and sorting for the similar purpose), risks taxonomical mis-assignment and, in the end, does not bring new findings or, at least, fails to stress their significance in the text. Also, I would suggest revisiting the terminology used and be careful not to use too strong statements when referring to the putative phylogenetic affiliations (or “origins” which I believe is quite a misleading term to use here).
A significant, yet I believe easily mendable flaw of the study is, that the comparison of the different independent studies fails to mention which strains of E. gracilis were used and reflect their possible genetic heterogeneity (which is in fact conceivable, given how fast at least some genetic elements, for instance introns, of E. gracilis mutate) in the discussion.
I also noted that especially the Methods section of the article is very detailed, to the degree of being quite unnecessarily wordy and repetitive at times and would in my opinion benefit greatly from pruning down the text or moving some of the lengthier descriptions (for example of the particular settings of bioinformatics tools used in cases when their value is specifically mentioned as unusual and/or crucial) to the supplemental files.
Here are my comments to concrete passages in the text.
Line 11: I think “is thought to be” is an unnecessarily weak statement
Line 16: text mentions five studies later; typo?
Line 30: the listing of the possible nutritional modes as “A, B, or C” somewhat obscures the fact, that the organism switches between and combines them readily; also Euglena is not really autotrophic, but rather mixotrophic in its photosynthetic mode
Line 42: the order of the sentence is a bit confusing, I think “It has been shown that many subunits previously considered exclusive to kinetoplastids are shared with E. gracilis and therefore (…)” communicates the meaning better.
Line 76: maybe mention where else can this polysaccharide be found?
Line 77: I believe the anti-tumoral activity is quite dubious, I would weaken the statement.
Lines 108-109: the description of the query search is a bit too detailed and slightly amusing.
Line 114: Unclear whether it implies those five studies were the only ones to employ Illumina? Please clarify, and if not the case, explain why keep these five and not the others, how was the decision made.
Lines 150-155: It does not make much sense to describe the hardware if its direct implication (for example the computational time) is discussed.
Line 160: Please clarify if these were true biological replicates (i.e. two different batches of culture) or technical replicates of the same sample.
Line 161: What does “separately but similarly” mean? I’m confused by the “similarity” claim (as opposed to “the same way” which would seem as an obvious baseline in bioinformatic analysis).
Line 163: “variety of metrics” is a bit vague, please state which metrics or simply state “(…) assess the quality of the data.” with reference to the program.
Line 170: I don’t understand the sentence, both the meaning of the “ergo” statement, and what is meant by the “visual inspection”?
Lines 193-194: long and a bit clumsy formulation, how about “(…) with e-value threshold of 1E-50 and SI threshold of 90%”?
Lines 197-210: Were these really the most abundant? Not other excavates or algae/plants? I would expect more explanation of this – and it is actually provided later in the text where it is discussed in more detail and context. I.e. why not omit this part or make it much briefer, and refer to the concrete organisms etc. in results/discussion?
Line 211: I would move the previous mention of EvidentialGene (in chapter 2.2.2) to this chapter.
Line 233: The exclusion of unpaired and singleton reads is mentioned for the first time here. Were they not excluded in the part of the workflow described in chapter 2.2.1 too?
Line 236-237: I find having a separate sentence to describe the program’s function unnecessary.
Line 238: Which custom scripts are meant? I understand they are not included in the supplemental files?
Lines 242-246: The explanation is not necessary, I would simply state that completeness was assessed by BUSCO using both prok and euk datasets.
Lines 250-252: I’m not sure I understand. Does the order in which the transcriptomes were tested make a difference?
Lines 252-261: The whole paragraph is quite wordy, please consider rephrasing it in a briefer way.
Lines 265-267 and 269-270: The explanations are not needed.
Line 286: Taxonomic affiliation analysis; i.e. a missing word?
Line 321: It is previously mentioned that the SVA correction was abandoned, does this mean it was used in some but not other cases? Couldn’t this introduce bias?
Lines 323-332: Would benefit from rephrasing and making the text briefer.
Lines 351-358: Worth rephrasing and shortening, some sentences are not necessary.
Lines 378-388: I suggest moving this passage to Methods, albeit in much briefer phrasing.
Line 389: 2.6 billion raw reads?
Line 390: No need to state all three identifiers of the experiment, “our experiment” would be enough.
Lines 395-408: No reason to elaborate so much on the other studies, I would expect simple “We used X out of Y reads produced by [48], A out of B from [49]” etc.
Lines 429-438: Ideally move the aforementioned paragraph on animal contaminants here and ideally make it less wordy.
Lines 458-459: “..showed interesting results.” – filler-like introductory sentence.
Line 466: How do you explain this cohort of low GC < 500 nt sequences? Is this a common artifact/ contamination, i.e. is it common and obvious practice to discard it? Or is it simply to reduce outliers? Can we be sure that this is not of biological significance, i.e. genuine codon bias cause, for example, by aerobic/anaerobic conditions or evolution of the strain?
Lines 493-495: I don’t understand the significance of this sentence.
Lines 499-508: Quite a lot of description of what can essentially be read from the table, I would omit or significantly shorten it.
Table 3: Missing reference for GDJR01
Lines 531-533: Long sentence + already mentioned.
Table 4: The E vs GDJR01 mapping of 51.39% reads is quite and outlier value compared to the other percentages in the table. How do you explain this?
Lines 562: Might have been affected how exactly?
Line 646: Calling the third category “eukaryotic genes” sounds like the other groups are not eukaryotic, please use better term.
Line 660: Anything worth mentioning regarding the bacterial groups identified? Did any taxonomical/ecological group dominate? What about chlamydiae (sometimes noticed as prokaryotic “background” signal in plastid-bearing organisms)?
Line 670: If the text claims that the proportion of foreign genes is “striking” it should mention concrete numbers to demonstrate this.
Lines 678-683: The alphabetical ordering of the species is a bit chaotic, ordering based on taxonomy would make more sense; also, some of these are neither trypanosomatids nor green algae/plants as suggested in the introductory sentence; I would collapse the trypanosoma species into one entry, i.e. genus Trypanosoma (7), or maybe collapse the whole list into less categories, i.e. trypanosomatids (8), plants (#), green algae (#) etc.
Line 732: Does the “(it) traces back to euglenozoan host” refer to the central role of mitochondrion, or the mitochondrion itself?
Figure 5: The vertical axes are not labelled; the labels are quite small, I’m afraid there will be difficult to read even if the graphic takes a whole page. Please clarify in the legend what do the trees represent. Also, I suggest mentioning the color-coding in the legend as well; and possibly choosing a different set of abbreviations for the conditions than for the experiments; e.g. Mx instead of M for mixotrophic, etc.
Line 750: I understand the graphic is not based on the whole 2500 genes but concern only the 631, is this right?
Line 774-776: I believe the sentence’s word-order is not correct.
Lines 788-789: What could the down regulation of carotenoid pathway mean? Was there a difference in light conditions, for example?
Lines 792-795: Again, what could be the biological significance of this?
Line 830: Which previous cluster analysis does this refer to?
Line 817: I do not understand the “likely nuclear encoded” part. Organellar genomes are available, i.e. how can nuclear localization of a gene be only “likely”?
Figure 6: I’m afraid the labels are going to unreadable even if the figure ends up taking a whole page.
Line 850: Please substitute “interest” with “significance”.
Lines 892-899: This may be just a terminology misunderstanding but I believe neither shopping bag nor red carpet hypothesis does not prefer “true” endosymbiosis over kleptoplastidy scenarios. Also, worth discussion: are there other cases of kleptoplastidic euglenids? It is worth mentioning that in dinoflagellates, the group on which the shopping bag was postulated, kleptoplastidy is extremely common. Nevertheless, I commend the discussion of possible mizocytotic origin of euglenid plastid, and recent findings regarding Rapaza and Ross Sea Dinoflagellate.
Author Response
The authors present a meta-assembly based on five previously published transcriptomes and new sequencing data in order to somewhat remediate the absence of a satisfactory genomic dataset of E. gracilis by obtaining a transcriptome of so far the highest quality and degree of completeness. They subsequently use the dataset as reference to attempt a differential transcriptomic analysis using the reads generated in the previous studies and determine co-regulated gene clusters.
>>> This is correct except that there were only three other published transcriptomes. Here, we compare our own transcriptome (released in July 2020) to the last two ones (Yoshida et al. 2016 and Ebenezer et al. 2019), even if it also includes reads from O’Neill et al. (2015).
The attention dedicated to the various steps of the assembly process itself, such as multiple rounds of read-filtering, decontamination, pre-assemblies and general fine-tuning of the whole meta-assembly strategy (as far as I know not often employed in protist -omics) is extensive and impressive, so is the rigorous statistics behind the expression analysis; these are undoubtedly the main strength of this work.
>>> Thank you for this positive assessment of our work.
Unfortunately, the biological (functional and evolutionary) conclusions that could be drawn from the results of the gene expression analysis are quite limited and relatively unremarkable. This is largely due to the fact, that the admittedly complex modulation of E. gracilis proteome in response to different conditions takes place mainly at translational, not transcriptional level. This has been strongly suspected for some time and the authors were aware of it. As much as I find it important (albeit emotionally unrewarding) to demonstrate this fact in such comprehensive and definitive way, I would expect the authors to provide stronger reasoning for dedicating time to what is essentially re-attempting (albeit more rigorously and in depth) something that was previously noted to be relatively unlikely to yield very novel and conclusive findings, for example by pointing out flaws of the previous studies, discussing the potential room for improvement or citing studies where such a strategy paid off.
>>> We see your point. Basically, the answer is that our RNA-Seq data is quite old (generated between 2012 and 2014) and we have just been very slow at processing it. At the time, it was obtained to support a comparative proteomic study that should eventually be published in New Phytologist. In the revised manuscript, we have better specified that our reads are rather special because they were sequenced from DSN-normalized total RNA samples (i.e., without fishing out mRNAs using their poly-A tail). Please also refer to our answers to reviewer 1’s comments for more background about these two aspects.
While the inter-study gene expression analysis does not bring conclusive results, the network of co-regulated gene clusters connects at least some of the proteins into functionally distinct “hubs” and represents a useful resource at intra-study level and could be elaborated on more in the text, especially in regard to discussing the potential biological interpretation. Generally, the Discussion section of the paper is relatively descriptive and dry (a bit like if it were a continuation of Results) and would benefit from more attempts at interpretation of the data in relation to other, non-omics disciplines of Euglena studies (biochemistry, molecular biology, ecology etc.) and theorizing. I include suggestions/questions on the respective lines of the text, see below. For instance, I was very surprised by the profound difference in the proportion of trans-spliced transcripts identified in this study in comparison to previous works and frequently cited values and would be very interested to hear more of the authors opinion on what could be the story behind this phenomenon, how it could relate to what we know about euglenozoan trans-splicing, what could be the biological significance of it etc. Related to this, I also wonder what is the state of 5’-end completeness of the transcripts, especially compared to the previously published datasets.
>>> Again, we understand your frustration, but this manuscript is rather meant as a technical/methodological study that might be of use for colleagues trying to do meta-assemblies of protists sequencing data. Nevertheless, we have comprehensively addressed all your detailed comments below, including the one about the spliced-leader sequences. For this, we have performed a new analysis (see Figure S3) comparing the mapping of the SL-sequence across the three public transcriptomes, including our own. It revealed that our consensus transcriptome is not bad from this respect but that the one of Yoshida et al. (2016) is much better.
Lastly, the authors also performed a transcriptome-wide screening of phylogenetic origin/affiliation of the E. gracilis proteins which is unfortunately based on best BLAST hit approach only which is relatively superficial (especially compared to the analogous screening performed in some previous studies that employed large scale tree generation and sorting for the similar purpose), risks taxonomical mis-assignment and, in the end, does not bring new findings or, at least, fails to stress their significance in the text. Also, I would suggest revisiting the terminology used and be careful not to use too strong statements when referring to the putative phylogenetic affiliations (or “origins” which I believe is quite a misleading term to use here).
>>> Yes. We totally agree with your point. Our lab is specialized in phylogenomics and, clearly, we did not deploy our full arsenal of phylogenetic methods in this manuscript. The reason is twofold: 1) as you correctly pointed out, recent studies have done a good job in this respect, especially the ones of Ebenezer et al. (2019) and Novak Vanclova et al. (2020); 2) we are actively working at a large phylogenomic study trying to probe the amount of gene transfers among all complex algae, including Euglena, but this work clearly cannot be published in this technical manuscript. Regarding the word “origins”, we have searched for its mis-use and replaced it by a milder term, such as “affinities”, whenever required.
A significant, yet I believe easily mendable flaw of the study is, that the comparison of the different independent studies fails to mention which strains of E. gracilis were used and reflect their possible genetic heterogeneity (which is in fact conceivable, given how fast at least some genetic elements, for instance introns, of E. gracilis mutate) in the discussion.
>>> Excellent point. In fact, this issue has also been raised by reviewer 1. Please refer to our corresponding answers about this. Briefly, all studies used the “Z strain” except the one by O’Neill et al. (2015), which used a distinct strain.
I also noted that especially the Methods section of the article is very detailed, to the degree of being quite unnecessarily wordy and repetitive at times and would in my opinion benefit greatly from pruning down the text or moving some of the lengthier descriptions (for example of the particular settings of bioinformatics tools used in cases when their value is specifically mentioned as unusual and/or crucial) to the supplemental files.
>>> Following your advice, we have shortened the text in many places. We warmly thank you for your numerous editorial suggestions. Yet, the manuscript remains mostly technical, even in its Results and Discussion section. As aforementioned, it was meant that way.
“Minor” comments
Line 11: I think “is thought to be” is an unnecessarily weak statement
>>> Fair enough. The issue is that our own working hypothesis about Euglena is more on the kleptoplastidic side. Changed to “considered as”.
Line 16: text mentions five studies later; typo?
>>> Indeed. We were not counting our own study. Fixed.
Line 30: the listing of the possible nutritional modes as “A, B, or C” somewhat obscures the fact, that the organism switches between and combines them readily; also Euglena is not really autotrophic, but rather mixotrophic in its photosynthetic mode
>>> We agree with your comment. We modified the first sentence to avoid mentioning the nutritional modes, an aspect which is already developed further in the introduction: “... in the light (mixotrophic conditions), high concentration of organic carbon leads to a decrease in photosynthesis by repressing chlorophyll biosynthesis”.
>>> We slightly modified the latter sentence according to your comment: “E. gracilis can therefore exploit a variety of organic carbon sources, as well in the dark (heterotrophic conditions) as in the light (mixotrophic conditions), where high concentration of organic carbon leads to a decrease in photosynthesis by repressing chlorophyll biosynthesis, reflecting the fact that this organism switches between nutritional modes and combines them readily.”
Line 42: the order of the sentence is a bit confusing, I think “It has been shown that many subunits previously considered exclusive to kinetoplastids are shared with E. gracilis and therefore (…)” communicates the meaning better.
>>> Thank you. Fixed as suggested.
Line 76: maybe mention where else can this polysaccharide be found?
>>> Paramylon is specific to photosynthetic euglenoids and stored in their cytoplasm (see also Monfils, A. K.; Triemer, R. E.; Bellairs, E. F. (2011). "Characterization of paramylon morphological diversity in photosynthetic euglenoids (Euglenales, Euglenophyta)". Phycologia 50 (2): 156). This reference has been added and the sentence changed: “In photosynthetic euglenoids, carbon reserves are stored in the cytoplasm in the form of paramylon (β-1,3-glucan) (Calvayrac et al., 1981; Monfils et al., 2011), in place of the starch (α-1,4 and α-1,6-glucan) typical of the green line.”
Line 77: I believe the anti-tumoral activity is quite dubious, I would weaken the statement.
>>> OK. Changed to “has been reported to display some anti-tumoral activity”.
Lines 108-109: the description of the query search is a bit too detailed and slightly amusing.
>>> OK. We were very thorough… Simplified.
Line 114: Unclear whether it implies those five studies were the only ones to employ Illumina? Please clarify, and if not the case, explain why keep these five and not the others, how was the decision made.
>>> Granted, this was unclear. At the time of our meta-assembly, those five studies were indeed the only transcriptional Illumina datasets publicly available and exploitable for Euglena gracilis. We have reworked this part to address another of your comments below. We now explain why we discarded 4 datasets among the 9 that were available.
Lines 150-155: It does not make much sense to describe the hardware if its direct implication (for example the computational time) is discussed.
>>> Indeed. Sentences deleted.
Line 160: Please clarify if these were true biological replicates (i.e. two different batches of culture) or technical replicates of the same sample.
>>> We understand your question but we were not able to find the answer in the corresponding publications (Yoshida et al. 2016 and Ebenezer et al. 2019).
Line 161: What does “separately but similarly” mean? I’m confused by the “similarity” claim (as opposed to “the same way” which would seem as an obvious baseline in bioinformatic analysis).
>>> Indeed. Changed to “all samples were treated separately.”
Line 163: “variety of metrics” is a bit vague, please state which metrics or simply state “(…) assess the quality of the data.” with reference to the program.
>>> OK. Simplified according to the suggestion.
Line 170: I don’t understand the sentence, both the meaning of the “ergo” statement, and what is meant by the “visual inspection”?
>>> Indeed, not that clear. Changed to “Finally, read quality was re-assessed using FastQC, and the resulting plots visually compared with those obtained in the beginning to check the effect of the filtering procedure.”
Lines 193-194: long and a bit clumsy formulation, how about “(…) with e-value threshold of 1E-50 and SI threshold of 90%”?
>>> Thank you. Changed according to the suggestion.
Lines 197-210: Were these really the most abundant? Not other excavates or algae/plants? I would expect more explanation of this – and it is actually provided later in the text where it is discussed in more detail and context. I.e. why not omit this part or make it much briefer, and refer to the concrete organisms etc. in results/discussion?
>>> No, because these were BLASTN searches specifically targeting sequence contaminations at the nucleotide level. Combined to the high-identity thresholds, they were not meant to capture sequences from genuinely related organisms, such as excavates or free-living algae (for the plastid). However, we have deleted the genome list from this section and moved it to the legend of Figure S1. The corresponding text is now much shorter. Thank you for the suggestion.
Line 211: I would move the previous mention of EvidentialGene (in chapter 2.2.2) to this chapter.
>>> We see your point but we cannot comply here because EvidentialGene was actually used at two different steps: to condense the four transcriptome “replicates” per experiment (section 2.2.2) and to condense the five resulting assemblies into one consensus assembly (section 2.2.4). Nonetheless, we have simplified section 2.2.2 to avoid the apparent repetition and removed unnecessary details.
Line 233: The exclusion of unpaired and singleton reads is mentioned for the first time here. Were they not excluded in the part of the workflow described in chapter 2.2.1 too?
>>> Here we discuss read mapping, not read assembly. What we mean is that we only counted read pairs in our statistics. Singleton reads (e.g., due to the decontamination step) and read pairs for which only one read actually mapped were not considered.
Line 236-237: I find having a separate sentence to describe the program’s function unnecessary.
>>> OK. Sentence deleted.
Line 238: Which custom scripts are meant? I understand they are not included in the supplemental files?
>>> In the “Data Availability Statement”, we provide a link to a GItHub repository with the custom scripts developed during our study: https://github.com/microalgues/clustering.
Lines 242-246: The explanation is not necessary, I would simply state that completeness was assessed by BUSCO using both prok and euk datasets.
>>> OK. Fixed as suggested.
Lines 250-252: I’m not sure I understand. Does the order in which the transcriptomes were tested make a difference?
>>> No, not the order, but these are pairwise comparisons and we systematically used our own transcriptome as the reference because the output is not symmetric.
Lines 252-261: The whole paragraph is quite wordy, please consider rephrasing it in a briefer way.
>>> Done. Thank you.
Lines 265-267 and 269-270: The explanations are not needed.
>>> OK. Explanations deleted.
Line 286: Taxonomic affiliation analysis; i.e. a missing word?
>>> Changed to “Taxonomic affinities were determined based on …”
Line 321: It is previously mentioned that the SVA correction was abandoned, does this mean it was used in some but not other cases? Couldn’t this introduce bias?
>>> No, SVA was used only for the PCA plots in Figure S8. The sentence was deleted to avoid the confusion and we now mention the prcomp function earlier in the text, when introducing another set of PCA plots.
Lines 323-332: Would benefit from rephrasing and making the text briefer.
>>> Done. Thank you.
Lines 351-358: Worth rephrasing and shortening, some sentences are not necessary.
>>> Done. Thank you.
Lines 378-388: I suggest moving this passage to Methods, albeit in much briefer phrasing.
>>> Done. Thank you.
Line 389: 2.6 billion raw reads?
>>> Yes. Fixed.
Line 390: No need to state all three identifiers of the experiment, “our experiment” would be enough.
>>> OK. Done.
Lines 395-408: No reason to elaborate so much on the other studies, I would expect simple “We used X out of Y reads produced by [48], A out of B from [49]” etc.
>>> OK. We have considerably reduced the paragraph.
Lines 429-438: Ideally move the aforementioned paragraph on animal contaminants here and ideally make it less wordy.
>>> We have shortened the aforementioned paragraph (see our answer above). Regarding the paragraph you point here, we have slightly shortened it as well.
Lines 458-459: “..showed interesting results.” – filler-like introductory sentence.
>>> OK. Sentence deleted.
Line 466: How do you explain this cohort of low GC < 500 nt sequences? Is this a common artifact/ contamination, i.e. is it common and obvious practice to discard it? Or is it simply to reduce outliers? Can we be sure that this is not of biological significance, i.e. genuine codon bias cause, for example, by aerobic/anaerobic conditions or evolution of the strain?
>>> We could not explain this cohort but we consider it either as an artefact or a contamination. In this section of the manuscript, we are still in a process of quality control, and we prefer not to speculate too much. However, we have added the following sentence to make the reader aware of the possible issue: “We could not determine what the removed sequences were by similarity searches (data not shown). They might represent some sort of artefact, contamination, or even be the result of a specific feature of experiment PRJEB10085 (B), for example the sequencing of a different strain, i.e., E. gracilis var. saccharophila Klebs (SAG 1224/7a) [O'Neill et al. 2015], whereas the other four experiments all used the Z strain (SAG 1224-5/25).”
Lines 493-495: I don’t understand the significance of this sentence.
>>> Indeed. Useless precision. Sentence deleted.
Lines 499-508: Quite a lot of description of what can essentially be read from the table, I would omit or significantly shorten it.
>>> We have shortened some sentences but we prefer to keep the text here.
Table 3: Missing reference for GDJR01
>>> Thank you. Fixed.
Lines 531-533: Long sentence + already mentioned.
>>> OK. Simplified to: “Using BUSCO on our predicted proteins, we found that the consensus transcriptome contained 84.8% of complete eukaryotic orthologs and half of them were duplicated, while 10.6% were missing (Figure 2). In comparison, we estimated the completeness of GDJR01 …”
Table 4: The E vs GDJR01 mapping of 51.39% reads is quite and outlier value compared to the other percentages in the table. How do you explain this?
>>> We have now mentioned in the text that our own reads (E) are distinct in nature from the reads of the other studies (e.g., they are richer in non-mRNA sequences). This probably explains their lower mapping percentage, even after filtering out low-quality reads. Regarding the specifically low mapping to GDJR01, it is difficult to explain, notably because “rRNA transcripts” are present in all three transcriptomes (Archive 1), including GDJR01. We have introduced these ideas in the text but have no definitive answer.
Lines 562: Might have been affected how exactly?
>>> This sentence has been deleted because of the improved analysis of spliced-leader sequences in the three public transcriptomes (Figure S3).
Line 646: Calling the third category “eukaryotic genes” sounds like the other groups are not eukaryotic, please use better term.
>>> We see your point. However, we used the categorization of Ahmadinejad et al. (2007) on purpose: “by the sequence similarity criterion, the genome of E. gracilis is expected to be a hybrid composed of four main gene classes: (i) Euglena-specific genes, (ii) Kinetoplastida-specific genes, (iii) eukaryotic genes that are spread in other eukaryotes, and (iv) genes acquired during the secondary endosymbiosis.” To address this comment, we have added the following precision in the revised text: “… (ii) kinetoplastid-specific genes, (iii) eukaryotic genes (i.e., widespread in other eukaryotes) ...”
Line 660: Anything worth mentioning regarding the bacterial groups identified? Did any taxonomical/ecological group dominate? What about chlamydiae (sometimes noticed as prokaryotic “background” signal in plastid-bearing organisms)?
>>> We do not feel very comfortable to draw specific biological conclusions from similarity searches. Nevertheless, we have added two sentences to the text: “Bacterial groups account for 1690 transcripts (12%), among which the most prominent are proteobacteria (34% of bacteria) and cyanobacteria (212, 13%). Only 40 (2%) and 15 (0.9%) transcripts are affiliated to the PVC group or Chlamydiae, respectively [Cenci et al. 2017].”
Line 670: If the text claims that the proportion of foreign genes is “striking” it should mention concrete numbers to demonstrate this.
>>> Fair enough. As aforementioned, we have a large phylogenomic study underway about gene transfers among all complex algae. For now, we cannot back up this claim using unbiased publicly available numbers (for example, in Curtis et al. 2012, only algal genes were reported). Therefore, we have reworked the sentence as follows: “Similarly to other complex algae (e.g., cryptophytes and chlorarachniophytes [Curtis et al. 2012], ochrophytes and haptophytes [Read et al. 2013 and Dorrell et al. 2017]), E. gracilis transcriptomes show a heavily mixed ancestry in terms of gene donor lineages.”
Lines 678-683: The alphabetical ordering of the species is a bit chaotic, ordering based on taxonomy would make more sense; also, some of these are neither trypanosomatids nor green algae/plants as suggested in the introductory sentence; I would collapse the trypanosoma species into one entry, i.e. genus Trypanosoma (7), or maybe collapse the whole list into less categories, i.e. trypanosomatids (8), plants (#), green algae (#) etc.
>>> Indeed. We have revised this paragraph to make it shorter and more accurate. In particular, we have performed a final control analysis (not described in the text) that confirmed 23 false positives among the 64 remaining contaminations and did not return additional contaminated transcripts (to our great relief).
Line 732: Does the “(it) traces back to euglenozoan host” refer to the central role of mitochondrion, or the mitochondrion itself?
>>> The mitochondrion itself.
Figure 5: The vertical axes are not labelled; the labels are quite small, I’m afraid there will be difficult to read even if the graphic takes a whole page. Please clarify in the legend what do the trees represent. Also, I suggest mentioning the color-coding in the legend as well; and possibly choosing a different set of abbreviations for the conditions than for the experiments; e.g. Mx instead of M for mixotrophic, etc.
>>> Good point about the abbreviations. For the anoxic/fermentative condition (A), we have changed it to “F”, hence there will be no confusion with the experiment codes (A to E). Of course, Figure 6 was also changed for congruence. We have also updated the legends to clarify the meaning of the trees. However, we have not changed the color coding.
>>> Moreover, in the PDF that was generated for the original review, the quality of the figures was unfortunately poor and the labels barely readable. However, the quality of the final figures will be sufficient to read the labels.
Line 750: I understand the graphic is not based on the whole 2500 genes but concern only the 631, is this right?
>>> Yes. We have improved the figure legend for the sake of clarity: “Selected co-expression clusters computed on the 2500 most variable genes. Only the five clusters characterized by significantly overrepresented ontological terms (featuring 631 transcripts) are shown.”
Line 774-776: I believe the sentence’s word-order is not correct.
>>> Thank you. Fixed.
Lines 788-789: What could the down regulation of carotenoid pathway mean? Was there a difference in light conditions, for example?
>>> If your comment refers to “In study PRJNA289402 (D), ABC transporters, fatty acid and polyketide synthesis were more down-regulated than in the remaining studies”, it is not specifically the carotenoid pathway that is downregulated, and we have no explanation at this stage.
Lines 792-795: Again, what could be the biological significance of this?
>>> There are indeed possible links between nitrogen metabolism and photosynthesis documented in many photosynthetic organisms but, considering that we cannot identify the factor that lead to these specific changes in experiment D compared to other experiments, we do not want to speculate too much.
Line 830: Which previous cluster analysis does this refer to?
>>> We assume that your comment refers to: “This observation thus corroborates what has been observed for the previous cluster analysis” at original lines 810-811. We have deleted the sentence because it was indeed confusing.
Line 817: I do not understand the “likely nuclear encoded” part. Organellar genomes are available, i.e. how can nuclear localization of a gene be only “likely”?
>>> You are correct. The formulation was clumsy. The idea was to mitigate the possibly-surprising non-chlorophyte affiliation. Changed to: “Concretely, genes coding for light-harvesting complexes grouped together distantly from other chloroplastic components. These transcripts are nuclear-encoded and showed a taxonomic affinity to Streptophyta.”
Figure 6: I’m afraid the labels are going to unreadable even if the figure ends up taking a whole page.
>>> See our answer to a similar concern you expressed above.
Line 850: Please substitute “interest” with “significance”.
>>> Done.
Lines 892-899: This may be just a terminology misunderstanding but I believe neither shopping bag nor red carpet hypothesis does not prefer “true” endosymbiosis over kleptoplastidy scenarios. Also, worth discussion: are there other cases of kleptoplastidic euglenids? It is worth mentioning that in dinoflagellates, the group on which the shopping bag was postulated, kleptoplastidy is extremely common. Nevertheless, I commend the discussion of possible mizocytotic origin of euglenid plastid, and recent findings regarding Rapaza and Ross Sea Dinoflagellate.
>>> In our view, kleptoplastidy means that there is no symbiotic relationship: the alga is a prey that is killed when ingested by phagocytosis or myzocytosis. In contrast, in endosymbiotic scenarios, the alga as a whole eventually becomes the organelle. Of course, the final symbiont (“plastid in the making”) could benefit from the products of genes transferred to the nucleus from earlier symbionts or kleptoplastidic preys. From the “shopping bag” paper: “We can readily envisage a situation where a host forms a transient relationship in which the **endosymbiont** persists for a while, and is then lysed. The host might form a series of transient **symbioses** before a stable relationship is finally achieved, in which the resulting organelle is a chimaera of products from different predecessors.” From the “red carpet” paper: “We introduce here the ‘red carpet hypothesis’ to propose that these red algal genes, transferred to the host nucleus before and/or during the early steps of **endosymbiosis** with green algae, provided important plastid-related functions and acted as a sort of ‘red carpet’ to facilitate the subsequent adoption of the new green algal endosymbionts.” Hence, both these hypotheses have been originally designed for endosymbionts, but we completely agree with you that they work for kleptoplastids too. As explained in our conclusion, a conceptual difference is that kleptoplastidy would better explain the selective pressure for transferring genes to the nucleus (i.e., to service the kleptoplastids deprived of their original source of nuclear-encoded proteins). We now clearly explain this idea in the slightly revised Conclusions section. In our opinion, such a pressure would be less obvious for (autonomous) free-living algae continually re-ingested from the environment. Moreover, kleptoplastidy would avoid the “Matryoshka dolls”-like membrane inflation predicted by serial endosymbiotic models. In the Conclusions, we do not discuss this issue, which is more relevant for (complex red) CASH lineages, Instead, we simply link the 3-membrane configuration of the Euglena plastids with the myzocytosis-based ingestion of kleptoplastids (observed in Rapaza). Nevertheless, this is not such an original idea. In fact, we have added the original reference proposing just that (Schnepf and Deichgräber, 1984). Regarding the extent of kleptoplastidy in dinoflagellates vs. euglenids, we are not aware of additional examples in euglenids. However, we have added a sentence and a reference to the dual feeding mechanism in Peranema (Triemer, 1997) to better back up the occurrence of myzocytosis in euglenids. In any case, thank you very much for your knowledgeable and positive assessment on our work!
Reviewer 3 Report
This paper reports a new combined transcriptome of E. gracilis using all the published data and some new data. This adds nothing new to our knowledge of Euglena biology - most sections include the phrase "similar to results found by XXX". There is a great deal of explanations of bioinformatic analysis done, but there are no new conclusions. The overall conclusion that "Alas, as it now appears quite clearly, gene expression is mostly controlled at the post-transcriptional level in euglenozoans" indicates absolutely no comparative transcriptomes should ever be done in euglenozoans. De novo transcriptomes of different species or genera of euglenozoans are still valuable as they will show different capabilities, but any more transcriptomes of E. gracilis are a waste of resources.
The cluster patterns show that expression is driven by study not experiment. This shows there are absolutely no conclusions that can be drawn from this at all. Figures 5 and 6 are just illustrations of how this doesn't work. This entire study just shows that mRNA levels do not change with experiment, which is well known in euglenozoa (eg Yoshida 2016, Ebeneezer 2019, Montandon 1990 - doi:10.1093/nar/18.1.75).
The large amount of bioinformatic results are mostly irrelevant and could be reported as supplementary information. The headlines are that the new data has fewer ORFs and contributes little to the combined transcriptome, somehow losing 2% of transcripts from the contributing studies.
Why were only the 5 selected non Euglena organisms used to identify contaminants - why no Gram +ve bacteria, or Archae, or other fungi etc. This is an oddly narrow list. (three metazoa).
If mRNA was sequenced, how come in section 2.5 organellar encoded proteins are analysed - they should not be present in the mRNA fraction.
Minor comments:
P1L30 "facultative secondary green alga" is not correct - it implies it is only green or an algae sometimes - should be "facultatively anaerobic green alga"
P1L39 : odd line break at end.
P1L42: Euglena is not a good comparator for parasitism, Bodo saltans for instance is better.
P2L46: cell morphology was the first indicator that euglenids and trypanosoma were related, not nuclear rRNA.
P9 L 380: what does PRJEB21674 correspond to if not Euglena? Is this miss annotated?
P10 L389: "totalised" should be "totaled"
L395-396: Should probably read "O'Neill et all obtained" not "were declared to obtain".
Section 3.3.2 should probably come before section 3.3.1
Figure 4 - do the different colour mean anything?
Line 826: if mRNA degradation/stabilisation controlled expression level then presumably these would change transcript abundance, which is what is actually measured in comparative transcriptomics.
Author Response
This paper reports a new combined transcriptome of E. gracilis using all the published data and some new data. This adds nothing new to our knowledge of Euglena biology - most sections include the phrase "similar to results found by XXX". There is a great deal of explanations of bioinformatic analysis done, but there are no new conclusions. The overall conclusion that "Alas, as it now appears quite clearly, gene expression is mostly controlled at the post-transcriptional level in euglenozoans" indicates absolutely no comparative transcriptomes should ever be done in euglenozoans. De novo transcriptomes of different species or genera of euglenozoans are still valuable as they will show different capabilities, but any more transcriptomes of E. gracilis are a waste of resources.
>>> Agreed. The focus of our manuscript is on the technical side, considering the availability of two previously released transcriptomes. As underlined by reviewer 1, independent corroboration is not rewarding but useful to science in general. That is why we did not submit this methodologically sound but not biologically enlightening study to a high-profile journal. Nevertheless, as explained in our first answer to reviewer 1, our RNA-seq reads were generated without poly-A selection of the mRNAs (but after DSN normalization). So, we could have expected to bring something different, especially at the time where the initial experiments were performed (2012-2014; see Figure S9).
The cluster patterns show that expression is driven by study not experiment. This shows there are absolutely no conclusions that can be drawn from this at all. Figures 5 and 6 are just illustrations of how this doesn't work. This entire study just shows that mRNA levels do not change with experiment, which is well known in euglenozoa (eg Yoshida 2016, Ebeneezer 2019, Montandon 1990 - doi:10.1093/nar/18.1.75).
>>> You are absolutely correct. Yet, the two other reviewers have contrasting opinions about the issue: reviewer 1 does not think we have enough power to rule out transcriptional regulation, whereas reviewer 2 deems that we have demonstrated the lack of it quite convincingly in our work. Apparently, your position is more in line with reviewer 2 but this shows that issue may be worth mentioning once more in the literature. Please also see our answer to reviewer 1 about a comparison of abundance changes in proteins vs transcript for a subset of circa 120 genes.
The large amount of bioinformatic results are mostly irrelevant and could be reported as supplementary information. The headlines are that the new data has fewer ORFs and contributes little to the combined transcriptome, somehow losing 2% of transcripts from the contributing studies.
>>> Well. This is a bit harsh. We have nothing to answer to this comment, except maybe that there is no proof that the 2% lost sequences were genuine sequences. They might very well be rubbish sequences or isoforms from another strain (see our answer to reviewer 1 about O’Neill et al. (2015) having used another Euglena than “strain Z”). In fact, we “miss” only 518 sequences in comparison to the transcriptome of Yoshida et al. (2016). Finally, we have carried out additional analyses to assess the quality of our new transcriptome (see our answers the reviewers 1 and 2). We see that each of the three public transcriptomes (including our own) has its pluses and minuses.
Why were only the 5 selected non Euglena organisms used to identify contaminants - why no Gram +ve bacteria, or Archae, or other fungi etc. This is an oddly narrow list. (three metazoa).
>>> As explained in section 3.2.1 of the Results and Discussion (see also section 2.2.3 of Materials and Methods), these organisms were selected based on the outcome of unbiased BLASTN searches performed on “preliminary” individual transcriptome assemblies against NCBI GenBank (nt database). Logically enough, they correspond to known contaminants of our sequencing platform. Yet, we have performed an additional analysis (not reported in the manuscript) that confirmed that 41 contaminating sequences are present in the publicly available version of our transcriptome (Table S6).
If mRNA was sequenced, how come in section 2.5 organellar encoded proteins are analysed - they should not be present in the mRNA fraction.
>>> We did not focus on mRNA only. Instead, we used DSN normalisation allowing us to bypass poly-A selection. This was mentioned in the original submission but, following your comment, we now better insist on this originality of our reads, including in section 2.5.
Minor comments
P1L30 "facultative secondary green alga" is not correct - it implies it is only green or an algae sometimes - should be "facultatively anaerobic green alga"
>>> Indeed. This sentence was changed to: “Euglena gracilis is a secondary green alga that can grow in a wide variety of environments.” Its metabolism was better described later in the introduction and this was improved there too, to address reviewer 2’s comments.
P1L39 : odd line break at end.
>>> Thank you. Fixed.
P1L42: Euglena is not a good comparator for parasitism, Bodo saltans for instance is better.
>>> This is true. We added a sentence (and two new references) to mitigate the previous idea: “Yet, it is worth mentioning that free-living bodonids (e.g., Bodo saltans) are better comparators for parasitism [Jackson et al. 2008 BMC Genomics; Deschamps et al. 2011 MBE]”. We did not remove the whole idea because its relationship with trypanosomatids is one of the classic features often stated about Euglena gracilis.
P2L46: cell morphology was the first indicator that euglenids and trypanosoma were related, not nuclear rRNA.
>>> Thank you for this comment. We changed the corresponding sentence as follows: “The relationship between euglenids and kinetoplastids has been first proposed by T. Cavalier-Smith based on ultrastructural similarities (e.g., “mitochondrial cristae shaped like a flattened disc with a narrow neck”) [Cavalier-Smith, 1981 BioSystems], then supported by other lines of evidence, such as alignments of nuclear rRNA…” We prefer to cite TCS (1981) as the source here because it is meant as a classification of all eukaryotes, even if earlier papers may also have been suitable to address your comment.
P9 L 380: what does PRJEB21674 correspond to if not Euglena? Is this miss annotated?
>>> No. This BioProject corresponds to the OneKP Major Release (1000 Plant (1KP) Transcriptomes). It only includes a single SRA file for Euglena and it is not a Euglena gracilis. We modified the sentence to be more accurate: “... while PRJEB21674 only included a single euglenid sample (among 1179), yet labelled as “Euglena sp.”, …”
P10 L389: "totalised" should be "totaled"
>>> Thank you. Fixed.
L395-396: Should probably read "O'Neill et all obtained" not "were declared to obtain".
>>> Thank you. We have completely rewritten this paragraph, in answer to a comment of reviewer 2. It is now much briefer.
Section 3.3.2 should probably come before section 3.3.1
>>> We see your point, which is understandable. Our opinion is that, in the present case, it is rather a matter of taste. In particular, the same BLAST searches both serve for functional and taxonomic annotation, as explained at the beginning of section 3.3.1. At this stage, we prefer not to change the order of these two sections, both to avoid messing with cross-reference numbers to figures and tables and to limit track-change highlighting to really meaningful changes. Nevertheless, if you deem that such an inversion would really benefit the manuscript, we would of course happily comply.
Figure 4 - do the different colour mean anything?
>>> Each color corresponds to an ontological term (or a group of related ones). In the PDF that was generated for the original review, the quality of the figure was unfortunately poor and the terms barely readable. However, the quality of the final figure will be sufficient to read the labels directly on the figure.
Line 826: if mRNA degradation/stabilisation controlled expression level then presumably these would change transcript abundance, which is what is actually measured in comparative transcriptomics.
>>> This is a very good point. Thank you. We added a sentence (bracketed here) to underline this, even if we are aware that it does not completely fix the logical problem you raise: “In these parasites, gene regulation mostly occurs at the post-transcriptional level, through stabilization/degradation of mRNA molecules and control of mRNA translation (see Vesteg et al. 2019 for a recent review of the issue). *While the former mechanism should in principle change transcript abundance, the latter one might not be visible in comparative transcriptomics.* For example, Yoshida et al. (2016) observed little change at the transcriptomic level following anaerobic treatment …”
Round 2
Reviewer 3 Report
The authors have responded to my comments.